# Non-invasive optoacoustic imaging of glycogen-storage and muscle degeneration in late-onset Pompe disease

Lina Tan[1,2], Jana Zschüntzsch [3], Stefanie Meyer[3], Alica Stobbe[3], Hannah Bruex[3], Adrian P. Regensburger [1,2], Merle Claßen[1,2], Frauke Alves [4,5], Jörg Jüngert[1], Ulrich Rother[6], Yi Li [6], Vera Danko[1,2], Werner Lang [6], Matthias Türk[7], Sandy Schmidt[8], Matthias Vorgerd[9,10], Lara Schlaffke[9], Joachim Woelfle[1], Andreas Hahn[11], Alexander Mensch [12], Martin Winterholler[13], Regina Trollmann [1,14], Rafael Heiß [8], Alexandra L. Wagner[1,2,15,16], Roman Raming[1,2,16] & Ferdinand Knieling [1,2,16] ✉

Pompe disease (PD) is a rare autosomal recessive glycogen storage disorder that causes proximal muscle weakness and loss of respiratory function. While enzyme replacement therapy (ERT) is the only effective treatment, biomarkers for disease monitoring are scarce. Following ex vivo biomarker validation in phantom studies, we apply multispectral optoacoustic tomography (MSOT), a laser- and ultrasound-based non-invasive imaging approach, in a clinical trial (NCT05083806) to image the biceps muscles of 10 late-onset PD (LOPD) patients and 10 matched healthy controls. MSOT is compared with muscle magnetic resonance imaging (MRI), ultrasound, spirometry, muscle testing and quality of life scores. Next, results are validated in an independent LOPD patient cohort from a second clinical site. Our study demonstrates that MSOT enables imaging of subcellular disease pathology with increases in glycogen/water, collagen and lipid signals, providing higher sensitivity in detecting muscle degeneration than current methods. This translational approach suggests implementation in the complex care of these rare disease patients.

Pompe disease (PD) is a rare, autosomal-recessive metabolic myopathy caused by mutations in the gene that encodes for acid alpha-glucosidase (GAA)[1–3]. Regularly, GAA catalyzes the hydrolysis of glycogen to glucose, but in PD, its impaired activity results in a generalized build-up of glycogen in metabolic active organs, such as heart, muscle and liver[4,5]. The disease progress is variable in age of onset, severity of organ involvement and degree of myopathy[6]. There is a differentiation in infantile (IOPD) and late-onset (LOPD) forms based on cardiac involvement, age of onset and residual enzyme activity[7]. IOPD patients may have less than 1% GAA activity, therefore, quickly develop severe symptoms, such as cardiac involvement, resulting in a high mortality rate by year one if untreated[1,8]. Children and adults with LOPD have residual enzyme activity below 30%, leading to more slowly progressive limb-girdle type weakness and respiratory insufficiency[9,10]. Replacement therapies (ERT) are available, leading to a slower progression of cardiac and musculoskeletal involvement, prevention of deterioration of pulmonary function and increasing survival[11–14]. However, an early initiation of treatment may positively impact the overall treatment response[15].

The diagnosis of PD is usually established by confirmation of GAA deficiency, and confirmed by genetic testing[16,17]. Furthermore, PD patients require regular clinical follow-up monitoring, especially to assess the response to ERT[8,9,17–20]. While rapid determination of GAA in dried blood spots is possible, enzymatic analysis is unable to

discriminate between patients with PD and those individuals harboring pseudo deficiency mutations. In this regard, a tetraglucose oligomer (Glc(4)) in the urine and maltotetraose (Hex4) in plasma may hold promise as a biomarker to identify PD patients from individuals harboring pseudo deficiency mutations[21] and even to assess response to ERT[22,23]. However, the interpretation of the values is not trivial and must be considered with respect to the individual age of the patient[18]. Therefore, follow-up is mostly ensured by clinical and functional tests, which are essentially dependent on the individual patient's active cooperation and performance[1]. More recently, magnetic resonance imaging (MRI) studies in LOPD demonstrated significant correlations between muscle involvement and function[24–26] or efficacy of enzyme replacement therapy and degree of lipomatous muscle alterations[27]. Moreover, it clearly identifies the proximal to distal involvement pattern of the disease[25]. Particularly in young patients, MRI has attributable risks for the requirement of sedation or difficult positioning for patients with respiratory impairment. Therefore, there is an unmet need for non-invasive techniques to better and more objectively assess disease involvement directly in the muscles of PD patients with the lowest burden possible. In this regard, multispectral optoacoustic tomography (MSOT) may be used to quantitatively image subcellular tissue composition and visualize disease-specific muscle changes[28–31]. MSOT applies the principle of "light in and sound out" through short-pulsed near-infrared laser emission and ultrasound detection, enabling it to retrieve deep-tissue information[32–37]. The mechanism of thermal expansion-based optoacoustic imaging (OAI) is that optical energy is absorbed by tissue chromophores, such as hemoglobins, lipids, water or collagens, causing localized heating and expansion, generating detectable acoustic pressure waves[38]. In this work, we show that in LOPD patients, MSOT enables imaging of subcellular disease pathology with increases in glycogen/water, collagen and lipid signals, providing higher sensitivity to detect muscle degeneration than current methods.

## Results

### Phantom imaging reveals optoacoustic properties of glycogen

To identify a possible specific glycogen spectrum, we first aimed to determine whether we could visualize its spectrophotometric absorption. An increase in glycogen concentration did not change the specific peak of the photometric spectrum with the investigated wavelengths (Fig. 1A). We could observe an increase of absorption starting from 910 nm, as described in the literature for $H_2O$[39]. After subtracting $H_2O$ background, there is a flat spectrum curve remaining, with only an absolute absorption shift between 2% and 7% glycogen (Fig. 1B). In contrast, $D_2O$ does not show an increase of absorption within the observed wavelengths (Fig. 1C) and after subtraction of $D_2O$ background it showed similar curves as compared to $H_2O$ experiments (Fig. 1D), which indicates no specific spectrophotometric absorption of glycogen.

To examine the optoacoustic properties of glycogen, we used a preclinical imaging system designed for small animal imaging[40]. Using this setup, $H_2O$ and 2% glycogen in $H_2O$ were detected, while $D_2O$ and 2% glycogen in $D_2O$ had no specific signal increase (Fig. 1E). Next, we transferred this to a clinical imaging system[28,41], which detected higher signal values when glycogen was added into $H_2O$, especially starting at 910 nm and a pronounced peak at 980 nm (Fig. 1F). The $H_2O$, $D_2O$ and glycogen in $D_2O$ showed similar curve progression. To validate the transferability of our in vitro glycogen findings into an actual muscle, we developed an ex vivo muscle phantom. Using a 3D-printed mold filled with minced meat, we started diluting it in $H_2O$ and successively added glycogen. We increased the glycogen content in 50% steps in relation to the calculated concentration in pure meat and imaged it using the clinical system. Our comparison shows an increasingly higher signal over several single wavelengths (SWLs), including 830 nm, 850 nm, 980 nm, 1030 nm, and 1080 nm for increasing concentrations

of glycogen (Fig. 1G, H). This suggests that glycogen, although not detectable as a pure substance, offers an optoacoustic imaging target in the clinical setting due to its potentially high-water binding capacity.

### Implementation of a clinical trial to study MSOT imaging

As reported previously, MSOT was capable of characterizing muscular remodeling with high sensitivity in neuromuscular diseases[28,30,42]. Given our in vitro findings in glycogen models, we hypothesized that it is possible to rapidly quantify LOPD-specific muscle involvement already in less affected proximal muscle groups. To perform an in vivo study of the MSOT imaging approach, we implemented a clinical trial. After regulatory approval and prospective registration to the clinical trial register, we included a total $n = 10$ healthy volunteers (HV), which were gender and age-matched to $n = 10$ LOPD patients (Fig. 2A). Besides clinical standard assessment, all patients were imaged using MSOT (Fig. 2B) and the resulting data was processed (Fig. 2C). The mean age ± SD was 41.2 ± 14.2 years in HV compared to 40.6 ± 12.1 years in the LOPD patients' cohort. In each group, 5 [50%] subjects were females. Eight (80%) of the LOPD patients received enzyme replacement therapy (ERT) with Alglucosidase alpha (Myozyme®, Sanofi). Four (40%) of LOPD patients needed nocturnal ventilation support. To determine the degree of disease, all HV completed the Rasch-built Pompe-specific activity scale (R-Pact), Quick Motor Function Test (QMFT), Medical Research Council (MRC), Up and Go test (TUG). 9 HV completed 6-Min Walk test (6-MWT) (one patient was not able to perform 6-MWT). 10 LOPD patients performed the R-Pact, QMFT and MRC, 9 LOPD patients performed TUG and 6-MWT (one patient was not able to perform 6-MWT and TUG due to his disease progression). Overall scores were significantly lower in LOPD patients (matched $n = 10$ HV vs. $n = 10$ LOPD patients: R-Pact: 36.0 ± 0.0 vs. 29.3 ± 10.1, $P = 0.0156$; QMFT: 64.0 ± 0.0 vs. 46.7 ± 16.1, $P = 0.0078$; MRC: 180.0 ± 0.0 vs. 149.5 ± 35.3, $P = 0.0078$; TUG: 1.7 ± 0.4 s vs. 3.7 ± 3.0 s, $P = 0.1421$; 6-MWT: 671.3 ± 116.6 m vs. 560.3 ± 87.2 m, $P = 0.0736$). The standard assessments are given in Table 1.

### Standard ultrasound and magnetic resonance imaging in LOPD

Next, we investigated whether B-Mode ultrasound (US) or magnetic resonance imaging (MRI) was capable of determining biceps muscle involvement in PD (Fig. 3A). For US, a total of $n = 40$ independent scans of the biceps muscle of HV ($n = 20$) and LOPD patients ($n = 20$) were evaluated. In HV, all 20 (100%) muscles were rated normal by an experienced and certified clinical investigator. Given the variability of the clinical phenotype, blinding was not fully feasible for all patients. By comparison, only 13 (65%) of the LOPD patients' biceps scans were rated normal, with 7 (35%) showing an overall pathological rating (Table 2).

Next, a comparison of the greyscale level (GSL) of the muscles between the diseased and the healthy subjects was performed[43]. The values were retrieved for two different regions of interest. One ellipsoidal in the center of the muscle and one polygonal ROI outline the major proximal portion of the muscle. The total mean value per patient showed no significant difference both in elliptic (84.0 ± 9.5 arb.units vs. 93.4 ± 13.1 arb.units, $P = 0.17$, Fig. 3B) and polygonal ROIs (80.6 ± 7.8 arb.units vs. 90.2 ± 11.8 arb.units, $P = 0.1$, Fig. 3C).

As MRI fat fraction is supposedly one of the most sensitive methods to detect muscle involvement, all subjects underwent an MRI of the biceps muscle. Signal intensities of in-phase and fat-only images were used to estimate the fat fraction (FF) as a surrogate marker for muscle atrophy in accordance with the pathophysiology of PD. In total, $n = 20$ datasets of the respective right biceps of HV and LOPD patients were evaluated. Overall, no difference in the fat fraction as surrogate for muscle involvement was found between HV and LOPD patients (9.7 ± 2.2% vs. 14.3 ± 14.1%, $P = 0.38$, Fig. 3D). These findings are in accordance with clinical phenotype and previous studies showing a less pronounced muscle involvement in LOPD[25].

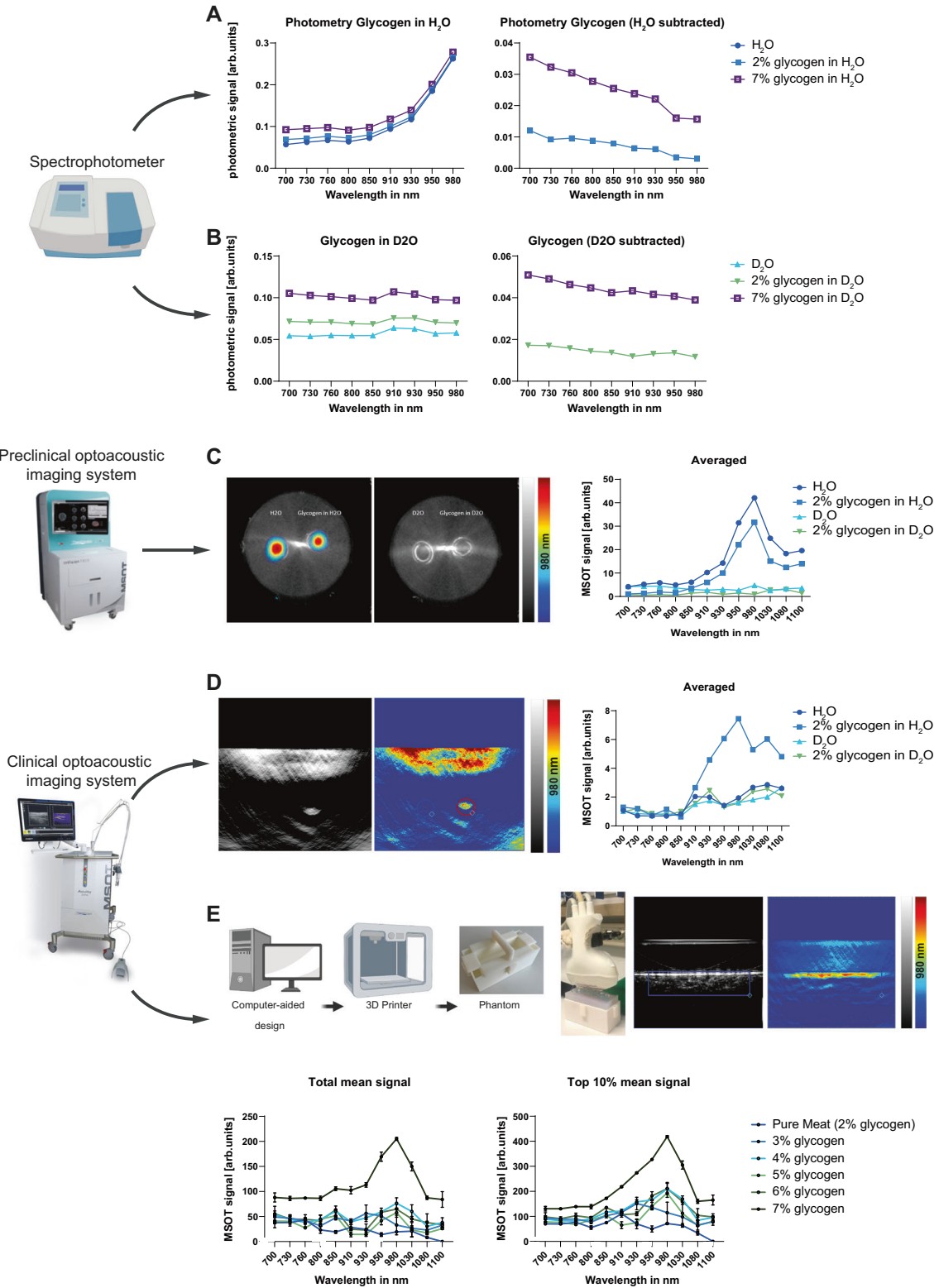

**Fig. 1 | Multimodal derivation of spectral information for glycogen reveals specific signatures for clinical imaging. A** Photometric absorption spectra between 700 and 980 nm for pure $H_2O$, and $H_2O$ with 2% and 7% glycogen, respectively (left). Photometric absorption spectra with subtracted $H_2O$ background (right). **B** Photometric absorption spectra between 700 and 980 nm for pure $D_2O$, and $D_2O$ with 2% and 7% glycogen, respectively (left). Photometric absorption spectra with subtracted $D_2O$ background (right). **C** Averaged optoacoustic signal in the preclinical imaging system from 700 to 1100 nm for pure $H_2O$, pure $D_2O$, and 2% glycogen in $H_2O$ and $D_2O$, respectively. **D** Averaged optoacoustic signal in the clinical imaging system from 700 to 1100 nm for pure $H_2O$, pure $D_2O$, and 2% glycogen in $H_2O$ and $D_2O$, respectively. **E** Averaged optoacoustic signal in the clinical imaging system from 700 to 1100 nm for pure minced meat and minced meat of the same origin with increasing glycogen concentrations. Values are given as mean values of scan data with negative signal intensities set to 0 or given as mean values of the top 10% of signal intensities per scan. The data represent one of two independent experiments with similar results. This figure was created with BioRender.com released under a Creative Commons Attribution-NonCommercial-NoDerivs 4.0 International license.

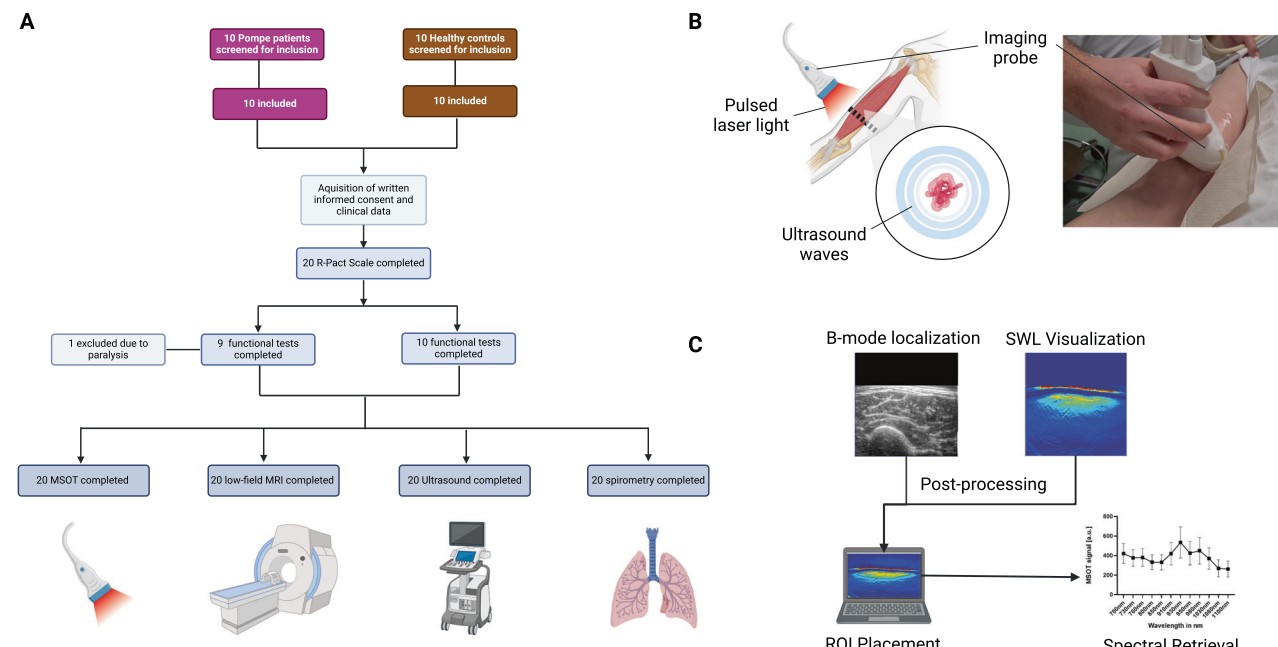

**Fig. 2 | Study flowchart imaging approach and quantification of scan results.**
**A** Consort flowchart diagram of the study. **B** Schematic and photographic representation of MSOT imaging approach. The imaging probe emitting pulsed laser light was held onto the distal third of the upper arm, scanning the biceps muscle.
**C** Localization of appropriate scan was performed on ultrasound B-mode images.

These were used to post-process optoacoustic spectral information. MSOT multispectral optoacoustic tomography, MRI magnetic resonance imaging, ROI region of interest, R-Pact Rasch-built Pompe-specific activity score. This figure was created with BioRender.com released under a Creative Commons Attribution-NonCommercial-NoDerivs 4.0 International license.

## MSOT enables visualization of biceps muscle involvement

For all subjects, MSOT imaging was completed, and data was post-processed using two independent scans of each biceps muscle for the final analysis. In total, $n = 80$ scans ($n = 40$ of HV, $n = 40$ of LOPD) were included, and signals, both SWLs and MSOT parameters (MSOT-derived lipid signal, $MSOT_{lip}$; MSOT-derived collagen signal, $MSOT_{col}$), were compared between groups. Figure 4A shows exemplary imaging results. Given the heterogeneity of the disease manifestation, each muscle was regarded as an individual data point. Therefore, $n = 20$ matched muscle regions were compared. The optoacoustic spectrum derived from 12 SWL showed overall higher values for PD patients compared to HV (Fig. 4B). $MSOT_{lip}$ provided better performance to distinguish HV from LOPD than clinical parameters (body mass index, BMI) or fat fraction derived by MRI (Fig. 4C). Spectral unmixing derived higher $MSOT_{col}$ ($1727 \pm 555.5$ arb.units vs. $2152 \pm 674.0$ arb.units, $P = 0.0029$), $MSOT_{lip}$ signals ($1267 \pm 356.6$ arb.units vs. $2713 \pm 1732$ arb.units, $P < 0.0001$), while in this approach SWL (800, 930, 980 nm) signals remained unchanged (Fig. 4D–H). However, the exact measure also depends on the proportion of the signal that is quantified (Supplementary Figs. 1–4). By separating patients with regard to disease severity based on QMFT, one can observe decrease of $MSOT_{col}$ and increasing $MSOT_{lip}$ quantification, possibly resembling the fibro-fatty degeneration of the muscle (Fig. 4I, J). $MSOT_{lip}$ demonstrated the strongest correlation to the other investigated clinical standard assessment (Fig. 4E).

## Proofing applicability and validity of MSOT using multicenter data

For better inter-device, -center and -examiner comparability, a similar investigation using an identical imaging approach on LOPD patients was conducted at a second center. This center produced $n = 3$ datasets from $n = 3$ individual LOPD patients, including duplicate scans of the right and left biceps muscle using the same imaging device. First, both centers independently analyzed the second center data and retrieved nearly identical results (Fig. 5A, B). Next, we compared the patient

cohorts ($n = 10$ vs. $n = 3$ LOPD patients) and compared the spectra (Fig. 5C, D). Most likely these are influenced by heterogeneity of the disease. When using spectral unmixed $MSOT_{col}$, we observed similar increased higher values in LOPD patients in both datasets (Fig. 5E). For $MSOT_{lip}$, we found larger differences in both centers (Fig. 5E).

## MSOT requires minimal scanning times

The investigation time for MRI depends on several factors, including the specific region of the body, patient-related considerations such as disabilities, patient cooperation, movement artefacts and repetition of imaging sequences. For the right biceps muscle scan time for T1 sequence requires 07:28 min, for T2 sequence 05:28 min. Additional time is needed to localize and optimize the scanning region, resulting in a minimal total scan time of 13:32 min.

In contrast, MSOT exhibits significantly shorter scan times. On average, one MSOT scan takes approximately 10 s. For the purpose of our study, we took two scans per biceps muscle This notable difference in investigation time highlights the advantage of MSOT in clinical use.

## Discussion

The findings presented here provide a rationale for a novel, non-invasive, radiation-free, and easy-applicable imaging modality to visualize disease-specific muscle patterns in LOPD patients. From a clinical perspective, we demonstrate the feasibility of MSOT to detect tissue remodeling caused by glycogen and its related higher water content. In the clinical imaging setup, glycogen in $H_2O$ has higher MSOT signals and a pronounced peak at 980 nm compared to pure $H_2O$. Studying muscle-mimicking phantoms, we found similar optoacoustic signal behavior correlating to the concentration of glycogen. Considering these results, we hypothesize that the current clinical optoacoustic system most likely visualizes glycogen-bound water. Taking this into consideration, we retrieved increased lipid and collagen signals as a sign of muscle degeneration in clinical subjects—also likely affected by the high glycogen-bound water in the muscle tissue. In contrast to our MSOT imaging findings, clinical routine US or MRI

**Table 1 | Characteristics of LOPD patients and healthy volunteers**

|  | HV *n* = 10 | LOPD *n* = 10 |
|---|---|---|
| Female, (%) | 5 (50) | 5 (50) |
| Male (%) | 5 (50) | 5 (50) |
| Age, years | 41.2 ± 14.2 | 40.6 ± 12.1 |
| Weight, kg | 73.9 ± 11.9 | 65.0 ± 17.3 |
| Height, cm | 176.2 ± 0.1 | 172.3 ± 0.1 |
| BMI (kg/m$^2$) | 23.6 ± 2.2 | 21.6 ± 4.0 |
| Ambulatory (%) | 10 (100) | 9 (90) |
| ERT (%) | 0 (0) | 8 (80) |
| Nocturnal ventilation support | 0 (0) | 4 (40) |
| **Lung function** |  |  |
| FEV1 (%) | 100.8 ± 11.8 | 68.2 ± 23.5 |
| FVC (%) | 104.9 ± 9.7 | 62.6 ± 21.4 |
| **Functional testing** |  |  |
| R-PACT | 36.0 ± 0.0 | 29.1 ± 10.4 |
| QMFT | 64 ± 0.0 | 46.7 ± 16.1 |
| 6MWT (m) | 671.3 ± 116.6 | 560.3 ± 87.2 |
| TUAG (s) | 1.7 ± 0.4 | 3.7 ± 3.0 |
| **MRC** |  |  |
| **UB** |  |  |
| Proximal | 5.0 ± 0.0 | 4.3 ± 0.1 |
| Medial | 5.0 ± 0.0 | 4.5 ± 0.1 |
| Distal | 5.0 ± 0.0 | 4.7 ± 0.0 |
| **LW** |  |  |
| Proximal | 5.0 ± 0.0 | 3.9 ± 0.1 |
| Medial | 5.0 ± 0.0 | 4.1 ± 0.1 |
| Distal | 5.0 ± 0.0 | 4.5 ± 0.1 |

Values are mean ± standard deviation (SD).
*LOPD* late-onset Pompe disease, *HV* healthy volunteer, *BMI* body mass index, *ERT* enzyme replacement therapy, *FEV1* forced expiratory volume, *FVC* functional vital capacity, *R-Pact* Rasch-built Pompe-specific activity score, *QMFT* quick motor function test, *6MWT* 6-min walking test, *TUAG* timed up-and-go test, *MRC* Medical Research Council score, *UB* upper body, *LB* lower body, *SD* standard deviation.

techniques were not feasible to visualize similar features in LOPD patients. Moreover, MSOT imaging is easily transferable to a second clinical center providing comparable imaging results. This unpins the utility and potential for MSOT for future non-invasive treatment monitoring in PD patients.

Currently, the first-line treatment in PD (IOPD and LOPD) is enzyme replacement therapy (ERT). In order to increase uptake, it is glycosylated with mannose-6-phosphate (M6P), which binds to cation-independent mannose-6-phosphate receptors (CIMPR) on cell membranes[44–46]. This allows affected tissue cells to take up the exogenous enzyme and increase their capability to degrade glycogen, subsequently clearing the excessive glycogen storages[11,47]. However, the affection of endocytic and autophagic pathways and the development of GAA-antibodies have a significant impact on trafficking and processing of ERT and subsequent therapeutic response[48]. As a consequence, the majority of patients show improvement or stabilization of respiratory and muscular functions, with more prominent effects at earlier stages of disease[49,50]. Laboratory biomarkers, including the assessment of GAA activity in dried blood spots or the analysis of Glc4 and Hex4, are applicable but not widely available or standardized across different centers[21–23]. Common tools to monitor PD patients are function-related, rely on active cooperation, may show a learning effect, and thus may not be sufficiently objective enough[51–53]. Moreover, therapeutic monitoring is currently becoming even more important because, in addition to new ERTs[47,54], genetic interventions

for PD patients are also on the horizon[55,56]. Second, early identification of LOPD patients in newborn screening programs may require more precise phenotyping[57].

Imaging technologies may offer the advantage of directly assessing changes in muscle structure and composition. The quantification of intramuscular fat or fat fraction (FF) has been shown to provide a solid correlation with muscle function and clinical outcomes in adults[24,25,58–60]. Less consistent results, as demonstrated in our study, facilitated the development of more sophisticated muscle MRI sequences, including diffusion tensor imaging[26] or glycogen spectroscopy[61,62]. Even without the presence of significantly increased fatty infiltration, these techniques can depict structural changes[26]. However, MRI may have several disadvantages, such as high costs, limited availability, lack of (inter-center) protocol standardization, and long examination time coupled with strict positioning of the patient. With progress in standardization and further clinical translation[63,64] as shown in this study, MSOT may enable easy-applicable imaging phenotyping of PD patients. The capability of MSOT has already been explored in gastroenterology[41,65–68], rheumatology[69], cardiovascular[70] and cancer medicine[71] or more specifically in muscle applications[42] such as Duchenne muscular dystrophy[28] and spinal muscular atrophy[30]. In this study, we assume that we were able to depict and quantify the muscle remodeling with localized glycogen accumulation, subsequently leading to fibro-fatty replacement and muscle atrophy. The spectral unmixed parameters, MSOT$_{lip}$ and MSOT$_{col}$, support our hypothesis that MSOT is capable of detecting such processes in the affected muscle.

In conclusion, MSOT holds great potential to become a sensitive imaging technology for diagnosing, phenotyping and monitoring patients with LOPD. With the increasing importance and availability of gene therapy, early treatment is vital. The findings of this work suggest the implementation of MSOT imaging into the comprehensive and complex care of these rare disease patients and could possibly reduce other more invasive procedures in the future. The next step should be a follow-up study including pediatric patients (both with IOPD and LOPD).

Nevertheless, this study has several limitations to consider. It is constrained by a small and heterogeneous sample size, which is attributed to the rarity of PD patients and the diversity in its clinical manifestations. Furthermore, MSOT is influenced by optical absorbers such as melanin, making it suitable primarily for individuals with lighter skin color and holds a maximum penetration depth of approximately 2.5 cm restricting MSOT application in certain cases and regions of the body. Further work is needed to develop a PD-specific methodology to evaluate disease-specific muscle involvement at different stages of the disease that relate to clinical findings. Moreover, optoacoustic imaging systems are still a novel imaging approach that have to be further improved and standardized[64]. Unmixing algorithms, reconstruction technique and analysis have to be enhanced to improve the quality of data[64,72].

## Methods
### Study design and subjects
A prospective, monocentric clinical study was conducted after receiving approval from the local ethics committee of the University Hospital Erlangen (UHE), Germany (reference: 21-238_1-B) and registration at clinicaltrials.gov (ID NCT05083806). This trial was performed according to the Declaration of Helsinki, and all subjects provided written informed consent. All investigations were performed at a single visit per participant. Inclusion criteria were confirmed Pompe disease (PD) independent from current therapy with an age over 18 years. Exclusion criteria were pregnancy, tattoo in skin area to be examined, contraindications for MSOT, MRI and for healthy volunteers (HV) any signs of myopathy. All participants were investigated between May 17, 2022, and March 30, 2023. A compensation of

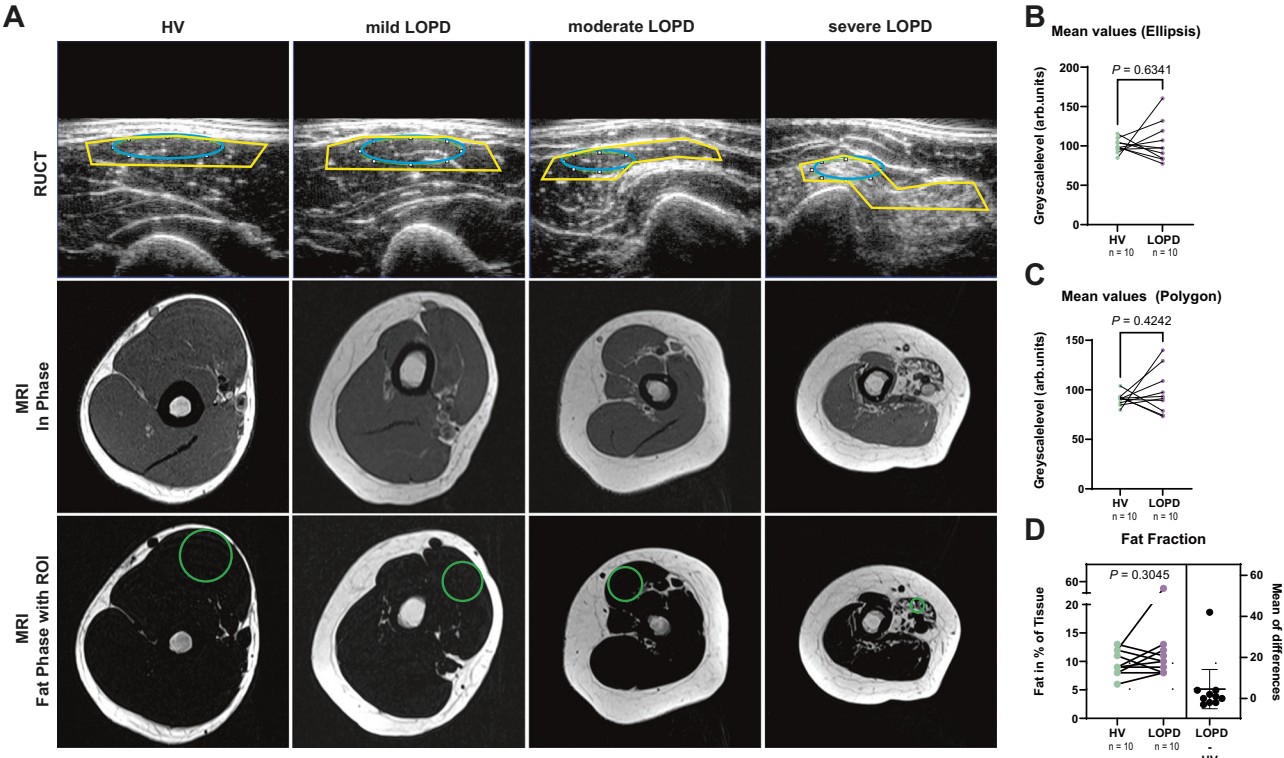

**Fig. 3 | Standard muscle imaging by ultrasound and magnetic resonance imaging does not show discernable differences in biceps muscles of LOPD patients.** **A** Ultrasound images (top row) and In-Phase (middle row) and Fat-Phase (bottom-row) MRI of the biceps muscle. From left to right, HV, mildly, moderately and severely affected LOPD patients. Elliptic (blue) and polygonal (yellow) ROI used in RUCT images and circular ROI used in Fat-Phase MRI for quantification. **B** Mean GSL values of matched HV vs. LOPD patients using an elliptic ROI. **C** Mean GSL values of matched HV vs. LOPD patients using a polygonal ROI. Each independent muscle region was scanned twice. Results represent 80 datasets from $n = 40$ independent biceps muscle regions ($n = 20$ HV/$n = 20$ LOPD) in $n = 20$ biologically independent subjects ($n = 10$ HV and $n = 10$ patients with LOPD). Each filled dot represents one MSOT signal per mean biceps muscle region (4 datasets from $n = 2$ independent independent muscle regions from one biologically independent subject). HV are represented with green and LOPD patients with violet dots. **D** ROIs in MRI images were manually placed in transversal slices of the right biceps brachii muscle corresponding to the position of MSOT evaluation. Results represent 20 datasets from

$n = 20$ independent biceps muscle regions ($n = 10$ HV/$n = 10$ LOPD) in $n = 20$ biologically independent subjects ($n = 10$ HV and $n = 10$ patients with LOPD). Each filled circle represents the fat fraction in percent per tissue signal per mean right biceps muscle (1 dataset from $n = 1$ independent muscle region from one biologically independent subject). HV are represented with green and LOPD patients with violet dots. To display differences in fat fractions, mean values from HV were subtracted from LOPD patients. One black dot represents one calculated ratio. Confidence intervals represent 95% CI ranging from −4.957 to 14.16, effect size ($R^2$) 0.1164, mean of differences (LOPD − HV) 4.6, SD of differences 13.36, SEM of differences 4.225. Two-tailed dependent samples $t$-tests (matched for age and sex) were used for statistical analysis. If the assumption of normal distribution was violated, a Wilcoxon signed-rank was used. $P \leq 0.05$ was considered statistically significant. HV healthy volunteer, LOPD late-onset Pompe disease patient, ROI region of interest, RUCT reflected ultrasound computed tomography, MRI magnetic resonance imaging.

300 Euro/participant was granted. The gender of LOPD patients and their matched controls was self-reported. The mean age ± SD was 41.2 ± 14.2 years in HV compared to 40.6 ± 12.1 years in the LOPD patients' cohort. In each group, 5 [50%] subjects were self-reported females.

For the inter-device, -center and -examiner comparison, the second site received an approval from the local ethics committee of the University Medical Center Göttingen (UMG), Germany (reference2/7/22). The inclusion and exclusion criteria followed the original approval of the University Hospital Erlangen.

### Phantom development and imaging
**Spectrophotometer imaging.** Absorption spectrum was measured every 10 nm in the range of 680–980 nm using the spectrophotometer SpectraMaxM2e (Molecular Devises, San Jose, USA). Six samples ($H_2O$, 2% glycogen in $H_2O$, 7% glycogen in $H_2O$, $D_2O$, 2% glycogen in $D_2O$, and 7% glycogen in $D_2O$) were measured separately.

**In vitro phantom imaging studies.** For in vitro imaging experiments, a phantom was custom-built by dissolving 2% agarose (Biozym LE

Agarose, Biozym Scientific GmbH, Hessisch Oldendorf, Germany) in heated $D_2O$ (Euriso-Top, St-Aubin Cedex, France). The warm fluid was filled into a cylindrical mold leaving two wells, each of 3 mm in diameter. After cooling down, the phantom solidified, leaving two separate holes within the phantom. Four samples: $H_2O$, $D_2O$, 2% glycogen (from bovine liver ≥85%, Sigma-Aldrich, St Louis, USA) in $H_2O$, and 2% glycogen in $D_2O$ were each mixed with 1.5% agarose at 37 °C. Each phantom was filled with two different samples, either with $H_2O$ and glycogen in $H_2O$ or $D_2O$ and glycogen in $D_2O$. Before injecting the warm fluid into the phantom holes, the phantom was placed on ice. This way, the warm and fluid samples cooled down and solidified quickly as they were injected. Next, imaging was performed from 660 nm to 1300 nm in 5 nm steps using a preclinical MSOT inVision Echo system (iThera Scientific, Munich, Germany). This system is also capable of acquiring interleaved US images for the coregistration of imaging data. Data was analyzed using viewMSOT software (version 4.1, iThera Medical GmbH, Munich, Germany). A region of interest (ROI) was drawn around the sample inclusions using the US image for local guidance. Mean signal intensity in the ROI at each wavelength was quantified and then plotted into an optoacoustic spectrum. We transferred this setup to a clinical

**Table 2 | B-mode ultrasound results of biceps muscle regions**

|  | Ultrasound scoring | HV (n = 10 individuals, n = 20 scans) | LOPD (n = 10 individuals, n = 20 scans) |
|---|---|---|---|
| Echogenicity | Hypo-echogenic | 16 (80%) | 7 (35%) |
|  | Isoechogenic | 4 (20%) | 4 (20%) |
|  | Hyperechogenic | 0 | 9 (45%) |
| Muscle texture | Coarse-granular | 3 (15%) | 1 (5%) |
|  | Medium-granular | 3 (15%) | 4 (20%) |
|  | Fine-granular | 14 (70%) | 15 (75%) |
| Distribution pattern | Focal | 0 | 1 (5%) |
|  | Inhomogeneous | 4 (20%) | 6 (30%) |
|  | Homogeneous | 16 (80%) | 13 (65%) |
| Heckmatt scale | 1 | 20 (100%) | 18 (90%) |
|  | 2 | 0 | 2 (10%) |
|  | 3 | 0 | 0 |
|  | 4 | 0 | 0 |
| Pathological | No | 20 (100%) | 13 (65%) |
|  | Yes | 0 | 7 (35%) |

*N* = 40 images (*n* = 20 HV/*n* = 20 LOPD) were evaluated for echo intensity, muscle texture, distribution pattern, Heckmatt scale and pathological rating. The investigator (J.J.) assessed echogenicity (hypo-/iso-/hyperechogenic), muscle texture (coarse-/medium-/fine-granular), distribution pattern (focal/inhomogeneous/homogeneous) and Heckmatt scale (grade 1–4: 1 = normal muscle echo, 2 = increased muscle echo while bone is still distinct, 3 = increased muscle echo and reduced bone echo, 4 = very strong muscle echo and loss of bone echo) in parallel to the examination. Additionally, the muscle was evaluated by the overall impression as healthy or pathological. Categorial variables are provided as numbers and percentages. *N* = 40 independent biceps scans (*n* = 20 HV/*n* = 20 LOPD) in *n* = 20 biologically independent subjects (*n* = 10 HV/ *n* = 10 LOPD).

*HV* healthy volunteers, *LOPD* late-onset Pompe disease.

MSOT Acuity Echo system (iThera Medical, Munich, Germany). An identical phantom was established using the same materials but filled into a box-shaped mold. Thus, we improved the contact area for the handheld probe. The four samples ($H_2O$, $D_2O$, 2% glycogen in $H_2O$, and 2% glycogen in $D_2O$) were processed and injected according to the steps above. Thereafter, the phantom was imaged from 660 nm to 1300 nm in 10 nm steps. The same examiner analyzed the data and drew the ROI as described above.

**Ex vivo muscle mimicking phantom.** To develop an ex vivo muscle-mimicking phantom, we used ground beef as a tissue substitute. Similar to the phantom described before, a custom-built phantom was established from 2% agarose (Biozym LE Agarose, Biozym Scientific GmbH, Hessisch Oldendorf, Germany) dissolved in distilled water, which was filled in a custom, 3D-printed mold. The mold was developed using Autodesk Fusion 360 (V2.0.14567, Autodesk GmbH, München, Germany) and a 3D printer (Form 2, Formlabs. Inc, Somerville, MA, USA) utilizing White Resin V4 (Formlabs. Inc, Somerville, MA, USA)[65]. The phantom main body measured 96*46*40 mm, which covered and sealed a recess of 76 × 26 × 20 mm. This remaining recess was subsequently filled with ground beef and glycogen at increasing concentrations. The initial weight of minced meat was 20 g (= 100%), which was diluted in 20 ml of $H_2O$. We assumed that the glycogen content of minced meat is equal to 2%, which equals 0.4 g (= 100%) glycogen in 20 g of meat. Next, glycogen was gradually added into $H_2O$, starting from 0.2 g (= 50%) up to 1.2 g (= 300%). Added to the 20 g minced meat, which we assumed included 0.4 g glycogen, this added up to a maximum glycogen content of 1.6 g, which was equivalent to 400% of the initial glycogen concentration of pure minced meat. For all experiments, the MSOT imaging probe (MSOT Acuity CE, iThera Medical, München, Germany) was secured, fixed in a bracket, and

coupled to the agarose phantom using transparent ultrasound gel (Aquasonic Clear, MDSS GmbH, Hannover, Germany).

## Study flow

The patients with diagnosed PD were compared to sex- and age-matched HV. All study participants underwent clinical standard assessments, ultrasound, MSOT and MRI imaging. For clinical standard assessment, all subjects completed a PD-specific questionnaire (R-PAct-Scale), motor function tests as well as a lung function testing. A co-registered Reflected-Ultrasound Computed Tomography (RUCT) provides US-like (B-mode) images and allows for guidance during MSOT imaging. MRI was performed of the right biceps in accordance with the previous marking. At the second MSOT site (UMG), the patients with PD received the MSOT imaging at the same anatomical localization following the UHE protocol.

## Clinical standard assessments

The Rasch-built Pompe-specific activity (R-PAct) scale is a patient-based questionnaire to specifically quantify the impact of PD on daily life and social participation[53]. Briefly, it is composed of 18 items of increasing difficulty (from "comb hair" to "running") assessed on a rating scale ranging from (0) "unable to perform" to (1) "able to perform, but with difficulty" to (2) "able to perform with no difficulty". The scale provides external validity, reliability and good discriminating ability. The Quick Motor Function Test (QMFT) comprised 16 items, requiring each subject to complete each item to their best ability and were rated on a scale from 0 (no muscle contraction) to 4 (normal movement), resulting in a maximum score of 64 points[51]. QMFT is sensitive for proximal muscle strength and differences in disease severity. As non-specific functional tests, we used MRC scale, timed-up-and-go-test and 6 min walking test. Muscle strength according to the MRC scale was evaluated bilaterally in the proximal regions of the upper and lower body (shoulder and hip flexion, extension, abduction, adduction, internal and external rotation), as well as in the medium (elbow and knee flexion, extension), and distal regions (hand, finger, feet and toe flexion, extension, abduction, adduction) of the body. A total score was obtained by adding up scores between 0 (no muscle contraction) to 5 (normal strength) of all 18 assessed muscle functions bilaterally, leading to a maximum of 180 points. For TUAG (Timed Up-and-Go Test), participants were seated on a stationary chair and timed from standing up to completing a single step[73]. The 6-min-walk test (6MWT) measures the distances covered in 6 min; the use of walking aids as needed was permitted[74].

## Lung function

Conventional spirometry was completed in seated position by all subjects to quantify lung function. Parameters of interest, such as VC (vital capacity) and FEV1 (forced expiratory volume in 1 s), were assessed considering age, sex and weight. Each patient underwent a minimum of three measurements, and an average value was calculated.

## B-mode ultrasound

All B-mode ultrasound images were acquired and analyzed by a single DEGUM-certified sonographer (J.J., German Society for Ultrasound in Medicine (DEGUM), level III sonographer) using the Mindray, Zonare ZS 3 (Zonare Medical System Inc, Mountain View). All muscles were evaluated for muscle texture (coarse-, medium-, fine-granuled), echogenicity (hyper-, hypo-, echogenic), distribution pattern (in-, homogeneous, focal) and Heckmatt scale (echogenicity of muscle and bone; grade 1–4: 1 = normal muscle echo, 2 = increased muscle echo while bone echo is still distinct, 3 = increased muscle echo and reduced bone echo, 4 = very strong muscle echo and complete loss of bone echo)[30,75,76].

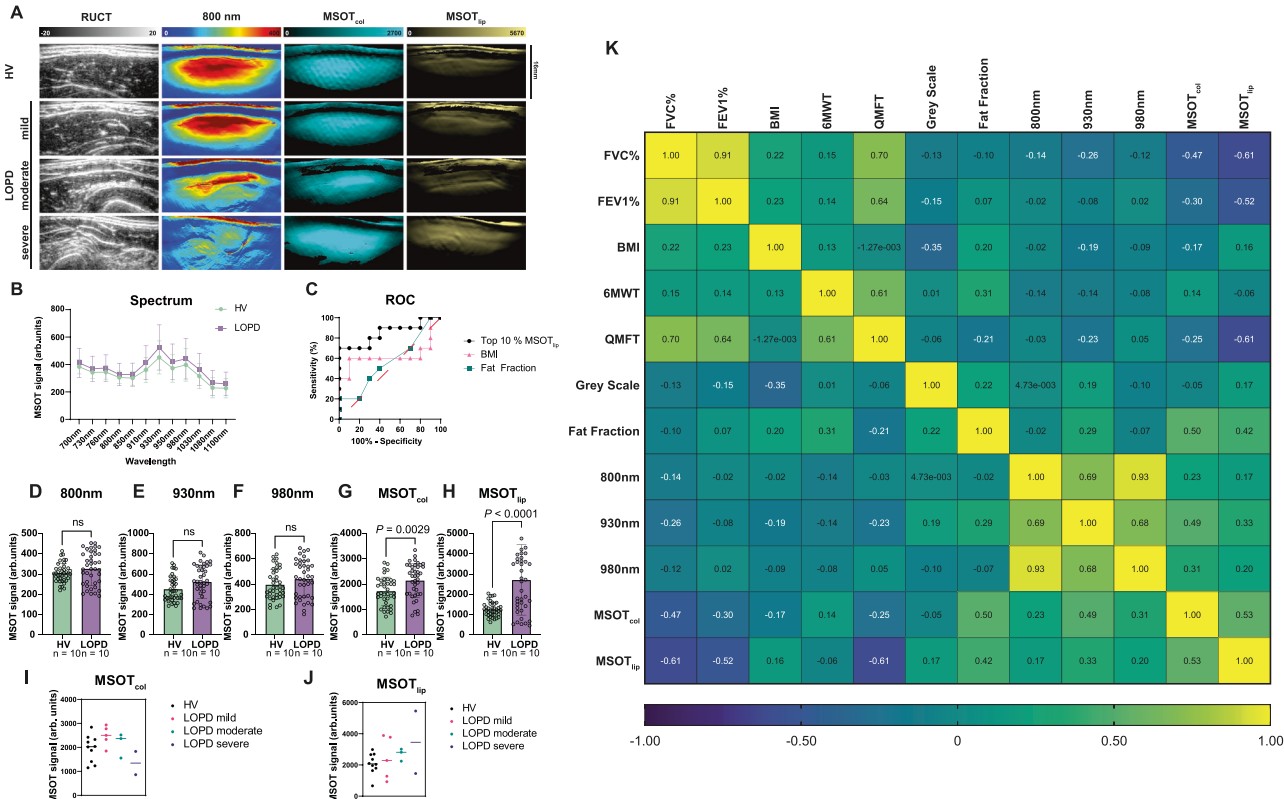

**Fig. 4 | MSOT quantification in human biceps muscles. A** From left to right: representative MSOT imaging quantification representing anatomic information (RUCT), unspecific tissue/muscle signal (SWL 800 nm), MSOT_col and MSOT_lip. Disease severity of HV vs. LOPD (mildly, moderately and severely) is increasing from top to bottom cases. **B** Comparison of MSOT spectral signal values of HV and LOPD patients from 700 to 1100 nm. Each dot represents the mean of a whole proband group (HV = green, LOPD = violet), bars represent 95% CI. Results represent 80 datasets from $n = 40$ independent biceps muscle regions ($n = 20$ HV/ $n = 20$ LOPD) in $n = 20$ biologically independent subjects ($n = 10$ HV and $n = 10$ patients with LOPD). **C** ROC Curve of Top 10% signals MSOT_lip, BMI values and MRI fat fraction values to distinguish HV and LOPD muscles. $n = 40$ independent muscle regions ($n = 20$ HV/ $n = 20$ LOPD) in $n = 20$ biologically independent subjects ($n = 10$ HV and $n = 10$ LOPD). Comparison of Top 10% of signal intensity for SWL 800 nm (**D**), 930 nm (**E**), 980 nm (**F**), MSOT_col (**G**), MSOT_lip (**H**) between HV and LOPD patients with individual scans as individual data points. Results represent 80 datasets from $n = 40$ independent biceps muscle regions ($n = 20$ HV/ $n = 20$ LOPD) in $n = 20$ biologically independent subjects ($n = 10$ HV and $n = 10$ LOPD). Each bar displays the mean of top 10% MSOT signal of the biceps muscle of a whole proband group with the error bars indicating SD (green bar/dots = HV and violet bar/dots = LOPD). MSOT signal comparison for different LOPD severity (HV =

black, mild = pink, green = moderate, severe = purple) for MSOT_col (**I**) and MSOT_lip (**J**). Results represent 80 datasets from $n = 40$ independent biceps muscle regions ($n = 20$ HV/ $n = 20$ LOPD) in $n = 20$ biologically independent subjects ($n = 10$ HV and $n = 10$ patients with LOPD). Each filled dot shows the mean of top 10% signal of the biceps muscle of different severity groups and HV (black dots = HV = QMFT = 64, pink dot = mild LOPD = QMFT 64–49, green dot = moderate LOPD = 48–33, purple dot = severe PD = 32–0). Statistical difference was tested with Welch's t-test. **K** Correlation matrix for maximum MSOT signal intensity of SWL 800 nm, 930 nm, 980 nm and MSOT_col and MSOT_lip correlated to reference clinical parameters including FVC%, FEV1%, BMI, 6MWT, QFMT, ultrasound greyscale levels, fat fraction. Correlations are indicated in the color range from highly negative (blue) to low negative/positive (green) to highly positive (yellow). Correlations are given by Spearman correlation coefficient (rs), two-tailed test. $P \leq 0.05$ was considered statistically significant. $n = 20$ biologically independent subjects ($n = 10$ HV/ $n =$ Confidence interval was 95% 10 patients with LOPD). HV healthy volunteer, LOPD late-onset Pompe disease patient, MSOT multispectral optoacoustic tomography, RUCT reflected ultrasound computed tomography, ROC receiver operating characteristic curve, FVC functional vital capacity, FEV1 forced expiratory volume, BMI body mass index, 6MWT 6-min walking test, QMFT quick motor function test.

## Magnetic resonance imaging acquisition and analysis

To derive the fat fraction, all participants underwent an MRI (MAGNETOM Free.Max, 0.55 T, Siemens Healthineers, Erlangen, Germany) of their right upper arm. A dedicated 6-channel surface coil was used (Siemens Healthineers, Germany). A transversal T1 turbo spin echo (TSE) Dixon sequence and a transversal T2 TSE Short-TI Inversion Recovery (STIR) were acquired. For MRI imaging analysis, manually drawn ROIs were placed in the biceps brachii muscle corresponding to the position of MSOT evaluation. Signal intensities of in-phase and fat-only images were used to estimate the fat fraction (FF). T2 TSE STIR images were read by a board-certificated radiologist with 10 years of experience in musculoskeletal imaging for the presence of intramuscular edema.

## Multispectral optoacoustic tomography

Imaging was performed by two MSOT-experienced examiners (A.L.W., R.R.) at UHE and two MSOT-experienced examiners (S.M.,

J.Z.) at UMG using at each site a different hybrid MSOT/RUCT (Reflected ultrasound computed tomography) imaging system (MSOT Acuity Echo, iThera Medical GmbH, Munich, Germany). As described before[28,30,41,42,66,77,78], a handheld 2D probe (center frequency: 4 MHz, field of view: 40 × 40 mm, 256 transducer elements, spatial resolution: 150 μm) was positioned at a 90° angle, and transparent ultrasound gel was used for coupling. Laser was set to a single wavelength (SWL) starting from 700 to 1210 nm (700 nm, 730 nm, 760 nm, 800 nm, 850 nm, 910 nm, 930 nm, 950 nm, 980 nm, 1030 nm, 1080 nm, 1100 nm, 1210 nm, Hb and HbO$_2$ spectrum) with a repetition rate of 25 Hz. Negative pixels were set to 0. Images required minimal motion of examiners, which was assisted by a motion bar provided by the software. A minimum of two images were taken per muscle. All participants wore safety goggles during the examination, adverse events were accordingly documented.

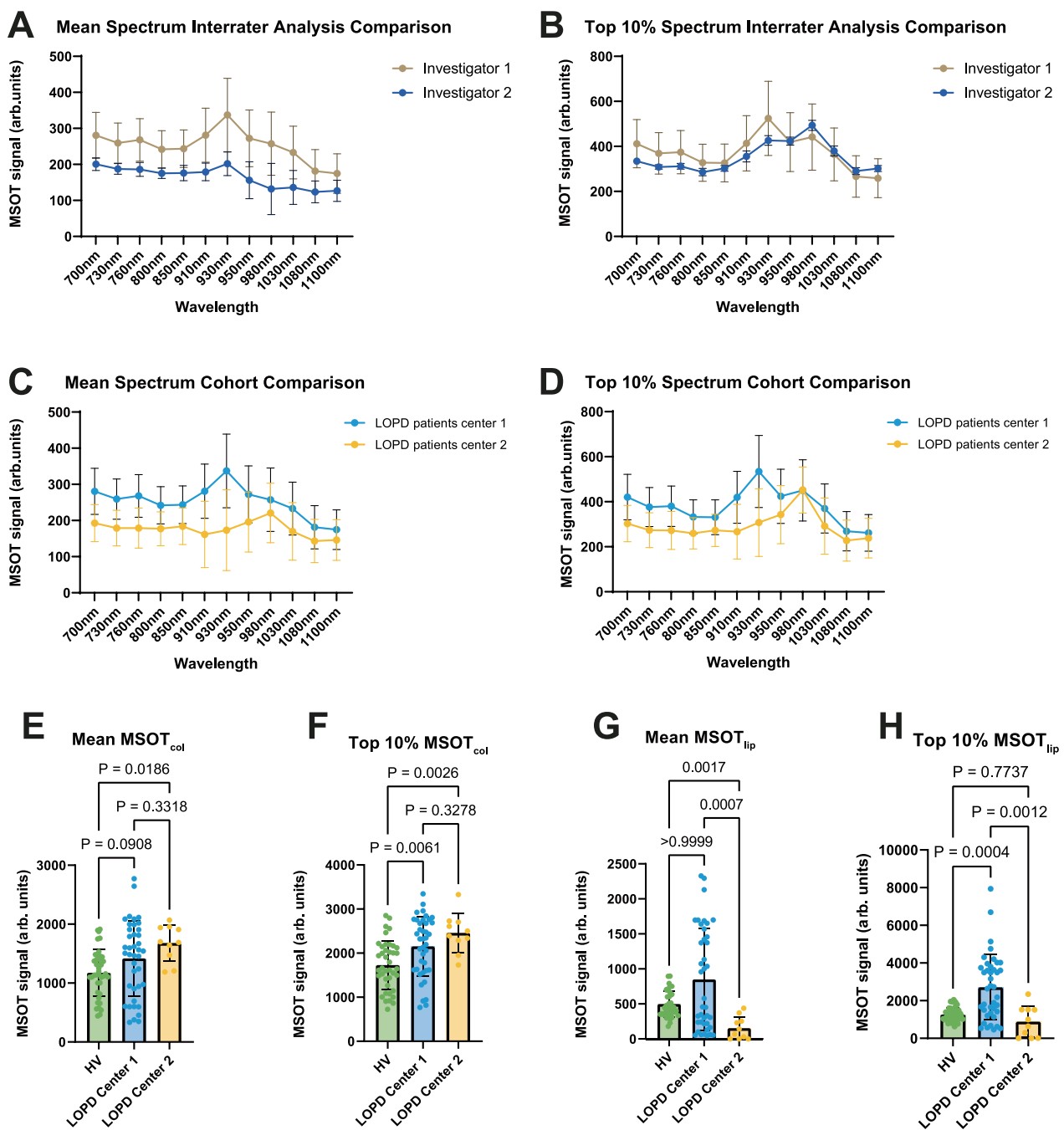

**Fig. 5 | Multicenter patient cohort comparison proves the applicability and validity of MSOT approach. A** Interrater analysis comparison of MSOT mean signals for two independent investigators (investigator 1, investigator 2) of two clinical centers (LOPD center 1, LOPD center 2) independently analyzed the same MSOT biceps muscle scans (10 datasets from $n = 6$ independent biceps muscle regions ($n = 3$ LOPD) in $n = 3$ biologically independent subjects). Data collected by center 2. **B** Interrater analysis comparison of MSOT top 10% signals for two independent investigators (investigator 1, investigator 2) of two clinical centers (LOPD center 1, LOPD center 2) independently analyzed the same MSOT biceps muscle scans (10 datasets from $n = 6$ independent biceps muscle regions ($n = 3$ LOPD) in $n = 3$ biologically independent subject). Data collected by center 2. Dual center comparison (center 1: $n = 10$ vs. center 2: $n = 3$) of mean (**C**) and top 10% (**D**) MSOT signals. Each filled blue circle displays the mean (**C**) and top 10% (**D**) of LOPD patients of center 1, each filled yellow circle displays the mean (**C**) and top 10% (**D**) of LOPD patients of center 2. Results of center 1 represent 80 datasets from $n = 40$ independent biceps muscle regions ($n = 20$ HV/$n = 20$ LOPD) in $n = 20$ biologically

independent subjects ($n = 10$ HV and $n = 10$ patients with LOPD). Results of center 2 represent 10 datasets from $n = 6$ independent biceps muscle regions ($n = 3$ LOPD) in $n = 3$ biologically independent subjects. Comparison of mean MSOT$_{col}$ (**E**), top 10% MSOT$_{col}$ (**F**), mean MSOT$_{lip}$ (**G**), and top 10% MSOT$_{lip}$ (**H**) between HV and LOPD patients of both centers. Green bar representing HV consists of 40 datasets from $n = 20$ independent biceps muscle regions of $n = 10$ biologically independent subjects. Blue bar representing LOPD Center 1 of 40 datasets from $n = 20$ independent biceps muscle regions of $n = 10$ biologically independent subjects, yellow bar representing center 2 consists of 10 datasets from $n = 6$ independent biceps muscle regions ($n = 3$ LOPD) of $n = 3$ biologically independent subjects. Ordinary one-way ANOVA was used for statistical analysis. If the assumption of normal distribution was violated, a Kruskal–Wallis test was used. Box plots are defined with a minimum at the 25th percentile, a maximum at the 75th percentile, center at the median value and whiskers at the minimal and maximal data points of each subgroup. MSOT multispectral optoacoustic tomography, HV healthy volunteer, LOPD late-onset Pompe disease patient.

## MSOT data analysis

MSOT Imaging data was transferred to a workstation and iLabs software (iThera Medical GmbH, Munich, Germany) was used for image analyses. Briefly, a polygonal region of interest (ROI) was defined and drawn centrally within the biceps muscle according to its B-mode US image. The ROI was placed directly under the muscle fascia, excluding any visible vessels. The size of the ROIs in the HV data was determined by the 800 nm optoacoustic signal and included only the high-intensity area to prevent false negative and underrated data. The PD patients' ROI size was determined by matching the area of the ROI to the one used with the matched HV (Max area Difference = 0.72 mm², Min area Difference = 0.02 mm², mean [SD] = 0.26 mm² [0.18]). Signal intensities of the above-named SWL were recorded as well as a spectral analysis to detect the MSOT parameters Hb, $HbO_2$, $MSOT_{collagen}$, and $MSOT_{lipid}$ was performed.

## Quantitative ultrasound greyscale scoring

The iLabs software (version 1.3.16, iThera Medical GmbH, Munich, Germany) was used to extract the MSOT/RUCT images as.png files. Specifically, only the RUCT images were chosen for the greyscale (GSL) analysis. As described before[43], the analysis was conducted using Fiji software, a distribution of the open-source ImageJ software (V2.1.0/ 1.53c). To facilitate this, polygonal ROIs were positioned beneath the muscle fascia. Afterward, the content within the ROIs was utilized to examine the GSL, wherein standardized mean, minimum, and maximum GSL values were quantified in arbitrary units. For further statistical analysis, only the mean values of GSL were used to enhance intermodal comparability.

## Statistical analysis

Continuous variables are given as means with SDs and categorical variables as numbers with percentages. A nonparametric Mann−Whitney test was used for unpaired comparisons. A nonparametric Wilcoxon test was used to assess differences in HV and paired PD patients. Adjusted P-values are reported. Prism 10, version 10.1.0 (GraphPad Software) was used for all statistical analyses. $P < 0.05$ was considered to indicate statistically significant difference in all analyses.

## Reporting summary

Further information on research design is available in the Nature Portfolio Reporting Summary linked to this article.

## Data availability

The raw (individual, identifiable patient) data are protected and are not available due to data privacy laws. Data sharing requests will be considered on a case-by-case basis. The processed pseudonymized imaging data can be accessed upon request and within the framework of legal regulations from the corresponding author (equivalent purposes to those for which the patients grant their consent to use the data). Access is granted directly after publication for 36 months. The contact is ki-forschung@uk-erlangen.de, and response to request will be provided within 4−6 weeks. The data will be available for 3 months. The remaining data of this study are provided in the Supplementary Information and Source Data file. The study protocols and the statistical analysis plan are provided with this manuscript in the Supplementary information file. Source data are provided with this paper.

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

## Acknowledgements

The research was supported by a research Grant from Sanofi Aventis (SGZ201912542) for A.L.W., R.R., R.T. and F.K. and an EU-IMI2 JU research grant (www.screen4care.eu) under the grant agreement No. 101034427 for V.D., S.M., J.Z., F.A., R.T., A.L.W., R.R. and F.K. The JU receives support from the European Union's Horizon 2020 research and innovation programme and European Federation of Pharmaceutical Industries and Association. Further support was provided by the European Research Council under the European Union Horizon H2020 program (ERC Starting Grant No. 101115742-IseeG) for F.K and an Else Kröner Excellence Fellowship from the Else-Kröner Fresenius Stiftung for A.P.R. A.L.W. was supported by the Rahel Hirsch Program scholarship from the Charité University Hospital Berlin. The present work was performed in (partial) fulfillment of the requirements for obtaining the degree "Dr. med." for L.T. The sponsors of the study had no influence on study design, data collection and analysis or manuscript writing.

## Author contributions

L.T., A.L.W., J.Z., A.P.R., V.D., R.R., and F. K. designed, performed experiments, and clinical studies. J.J., V.D., and S.S., S.M., A.S., and H.B. performed clinical studies. J.Z., M.T., U.R., Y.L., W.L., A.H., R.T., A.M., M.W., J.W., and R.H. provided essential support to the clinical study. U.R., F.A., M.V., and L.S. provided essential materials or technical expertise. L.T., R.R., A.L.W., M.C., and F.K. analyzed and interpreted the data. F.K. conceived and supervised the project. L.T., R.R., and F.K. wrote the first draft of the manuscript. All authors edited and approved the final draft.

## Funding

## Competing interests

A.P.R. and F.K. are co-inventors together with iThera Medical GmbH, Germany, on an EU patent application (EP 19 163 304.9) relating to a device and a method for analyzing optoacoustic data, an optoacoustic system and a computer program. F.K. is a member of the advisory board of iThera Medical GmbH, Munich, Germany. A.P.R. and F.K. received travel support from iThera Medical GmbH, Germany. A.P.R., A.L.W., and F.K. report travel support from Sanofi Aventis, Germany. A.P.R. and F.K. report lecture fees from Sanofi Genzyme. F.K. reports lecture fees from Siemens Healthcare GmbH. The other authors declare no competing interests.

## Additional information

[1]Department of Pediatrics and Adolescent Medicine, University Hospital Erlangen, Friedrich-Alexander-Universität (FAU) Erlangen-Nürnberg, Erlangen 91054, Germany. [2]Translational Pediatrics, Department of Pediatrics and Adolescent Medicine, University Hospital Erlangen, Friedrich-Alexander-Universität (FAU) Erlangen-Nürnberg, Erlangen 91054, Germany. [3]Neuromuscular Disease Research, Clinic for Neurology, University Medical Center Göttingen (UMG), Göttingen 37075, Germany. [4]Translational Molecular Imaging, Max-Planck Institute for Multidisciplinary Sciences (MPI-NAT), City Campus, Göttingen 37075, Germany. [5]Clinic for Haematology and Medical Oncology, Institute of Diagnostic and Interventional Radiology, University Medical Center Göttingen (UMG),

Göttingen 37075, Germany. [6]Department of Vascular Surgery, University Hospital Erlangen, Friedrich-Alexander-Universität (FAU) Erlangen-Nürnberg, Erlangen 91054, Germany. [7]Department of Neurology, University Hospital Erlangen, Friedrich-Alexander-Universität (FAU) Erlangen-Nürnberg, Erlangen 91054, Germany. [8]Institute of Radiology, University Hospital Erlangen, Friedrich-Alexander-Universität (FAU) Erlangen-Nürnberg, Erlangen 91054, Germany. [9]Department of Neurology, BG-University Hospital Bergmannsheil, Ruhr-University Bochum, 44789 Bochum, Germany. [10]Heimer Institute for Muscle Research, BG-University Hospital Bergmannsheil, 44789 Bochum, Germany. [11]Department of Child Neurology, Justus-Liebig-Universität Giessen, 35385 Giessen, Germany. [12]Department of Neurology, Martin-Luther-Universität Halle-Wittenberg, 06120 Halle (Saale), Germany. [13]Sana Krankenhaus Rummelsberg, 90489 Nuremberg/Schwarzenbruck, Germany. [14]Center for Social Pediatrics, University Hospital Erlangen: Friedrich-Alexander-Universität (FAU) Erlangen-Nürnberg, Erlangen 91054, Germany. [15]Department of Pediatric Neurology, Center for Chronically Sick Children, Charité Berlin, 13353 Berlin, Germany. [16]These authors contributed equally: Alexandra L. Wagner, Roman Raming, Ferdinand Knieling.  e-mail: ferdinand.knieling@uk-erlangen.de

