## [Peer Review File · Nature Communications]

REVIEWER COMMENTS

Reviewer #1 (Remarks to the Author):

Nature Communications Review

In "Non-invasive optoacoustic imaging of glycogen-storage and muscle degeneration in Late-onset Pompe disease," Tan and colleagues investigate whether multispectral optoacoustic tomography (MSOT) can be applied to measure glycogen accumulation in the muscle of patients with Late-onset Pompe Disease. MSOT is a modified ultrasound-like approach which emits near-infrared light and records sound waves emitted. It is non-invasive and the wavelengths emitted can be customized to measure molecules of interest with some precision.

Tan and colleagues began by testing in an agarose and water system if glycogen could be measured with spectrophotometer. They do not identify any signal from glycogen itself, but they are able to detect an increase in the absorption of water, which changes when the concentration of glycogen increases. They thus appear to be using an indirect measurement of glycogen accumulation (Figure 1A,B). They then move onto a preclinical imaging system, where again they can measure a difference in water content as a surrogate for increased glycogen (Figure 1C). Next, they move to a clinical-grade system where they can observe a more pronounced difference, again by measuring water content in the setting of increased glycogen (thus an indirect measurement). They further test the system by developing a model of human muscle by using minced ground beef and adding increasing concentrations of glycogen. In this system, they note a steady increase in signal, with a clearly separable peak when the concentration of glycogen increases to 7%.

With the preclinical data above, Tan and colleagues then move to test the MSOT technology in a small clinical trial of 10 patients with LOPD and 10 healthy control volunteers. The trial included MSOT, muscle MRI, muscle ultrasound, and physical therapy functional testing. Healthy controls were age and sex matched. Healthy controls were heavier (73.9k g vs 65 kg) and taller (176.2 vs 172.3 cm). Functional testing with the R-PACT, QMFT, 6MWT, and TUAG were consistent with typical scores seen in LOPD. Strength on the MRC scale was consistent with proximal > distal weakness typically observed in LOPD.

In Figure 3, results from muscle MRI and ultrasound are given and are consistent with results obtained in clinic and previous reports in LOPD. There is visual difference on MRI and conventional ultrasound with increased fibrofatty replacement, but this is not significant using the elliptic or polygonal ROIs selected by the investigators.

- What are the noteworthy results?

Tan et al. present MSOT as a new, non-invasive imaging technique as a biomarker in Pompe disease. They test the technique in a glycogen and water mixture and a phantom muscle model with additional glycogen added (minced ground beef with increasing normal glycogen). They find that they cannot directly measure glycogen with their MSOT technique, but they can measure altered water signal in the presence of glycogen.

In a clinical trial, they claim superiority to muscle MRI and muscle ultrasound, but from a practical level appear to show at least non-inferiority to these imaging modalities.

The MSOT technique has advantages compared to MRI – it does not require sedation for younger patients (though, note this study is in adult participants with mean age of 40 and not children).

The technique is faster than MRI or ultrasound as it can be performed in 10-20 seconds of imaging time.

- Will the work be of significance to the field and related fields? How does it compare to the

established literature? If the work is not original, please provide relevant references.

The work is original for glycogen storage disease. Similar work has been performed by coauthors on this paper for other neuromuscular disorders.

- Does the work support the conclusions and claims, or is additional evidence needed?

The claim appears valid that MSOT can distinguish LOPD from healthy volunteers, at least in this small study of 10 patients and 10 controls.

The authors claim that there are few biomarkers and these biomarkers can be unreliable --"Follow-up and monitoring are ensured by clinical and functional tests, which are essentially dependent on the individual patient's active cooperation and performance.1," however, they do not discuss other blood or urine based biomarkers, such as urine hex4, which is reliable and not influenced by the patient's cooperation of performance. Further discussion using updated guidelines for monitoring is recommended in the introduction and discussion.

The authors claim improved detection of disease compared to muscle MRI and ultrasound, however the correlation matrix in Figure 4 does not suggest a strong correlation between MSOT results and the various biomarkers and scores from the tests performed. It is also difficult to tell over the cohort if mild, moderate, and severe LOPD can be distinguished with MSOT. Will this biomarker be useful to trend overtime in an individual patient or does it just confirm disease is present? More sensitive and specific diagnostic tests already exist for Pompe Disease (enzyme activity testing and sequencing of GAA are widely available clinically and well validated).

The claim in the discussion that this MSOT will serve as a substitute for invasive procedures (biopsy) appears overstated. The MSOT presented here is not measuring glycogen, thus there may be a number of different conditions which alter water signal in MSOT – the results may be more related to fibrofatty replacement than glycogen accumulation. This will not replace biopsy for challenging diagnostic cases where staining for glycogen and characterization by light and electron microscopy give vastly more information than this technique.

- Are there any flaws in the data analysis, interpretation and conclusions? - Do these prohibit publication or require revision?

A concern is that, though MSOT is claimed to be "molecularly-sensitive," the data presented here show an indirect measurement of glycogen accumulation through alteration in water content. This seems to fall short of the direct measurement of glycogen, that the benefits of a molecularly-sensitive technique claims. There may be a number of alterations in patients with Pompe Disease or other neuromuscular disorders which alter water content or fatty replacement of muscle that is unrelated to glycogen accumulation. The major advance would be direct measurement of glycogen in muscle, which is not proven in this paper with this MSOT method.

I would recommend revision for the language around biomarkers that includes urine and blood biomarkers (Hex4, CK, etc).

From the data presented, I am not convinced that MSOT is superior to muscle MRI or ultrasound. It may be non-inferior. It would be helpful to visualize the results to show individual level data and separate by LOPD severity. Can severe LOPD be separated from moderate and mild LOPD? If there is no difference, how will this technique be helpful in a gene therapy or ERT trial? Both of these therapies slow, but do not fully eradicate, glycogen accumulation. Thus the patients will not go from diseased to normal. It is therefore critical for a biomarker to distinguish not only between healthy and disease, but between mild, moderate, and severe disease.

- Is the methodology sound? Does the work meet the expected standards in your field?

The methodology appears reasonable for the development of a preclinical model and for the clinical trial with appropriate functional testing and current imaging modalities (MRI and conventional ultrasound).

- Is there enough detail provided in the methods for the work to be reproduced?

Yes, the methods are appropriately detailed.

Reviewer #2 (Remarks to the Author):

In this manuscript, the authors applied multispectral optoacoustic tomography (MSOT) to image biceps muscles for late-onset Pompe disease patients. Their results demonstrated that MSOT enabled imaging of subcellular disease pathology with increases in glycogen/water, collagen and lipid signals providing higher sensitivity to detect muscle degeneration than current clinical and imaging methods. Overall, the work is very interesting, which would help to advance the translation of the new imaging technique MSOT. In addition, the manuscript is well presented with proper results and discussion, and the limitations have also been stated. Therefore, I would recommend its publication after the following minor issue is addressed:

In the last paragraph of the Discussion section, it is stated "MSOT holds great potential to become a sensitive biomarker for diagnosing, phenotyping and monitoring patients with LOPD". The technique MSOT itself cannot be a biomarker, but it can serve as an operable technique for diagnosing, phenotyping and monitoring patients with LOPD. Hence, This statement should be modified.

Reviewer #3 (Remarks to the Author):

This referee assessed only the statistical part and report of results.

Regarding these sections i have not particular comments. The analyses were simple and results adequately reported.

I have only a question. Is it possible to see the results in a multivariable analysis or isn't feasible.

Reviewer #1 (Remarks to the Author)

Comment 1

In “Non-invasive optoacoustic imaging of glycogen-storage and muscle degeneration in Late-onset Pompe disease,” Tan and colleagues investigate whether multispectral optoacoustic tomography (MSOT) can be applied to measure glycogen accumulation in the muscle of patients with Late-onset Pompe Disease. MSOT is a modified ultrasound-like approach which emits near-infrared light and records sound waves emitted. It is non-invasive and the wavelengths emitted can be customized to measure molecules of interest with some precision.

Tan and colleagues began by testing in an agarose and water system if glycogen could be measured with spectrophotometer. They do not identify any signal from glycogen itself, but they are able to detect an increase in the absorption of water, which changes when the concentration of glycogen increases. They thus appear to be using an indirect measurement of glycogen accumulation (Figure 1A,B). They then move onto a preclinical imaging system, where again they can measure a difference in water content as a surrogate for increased glycogen (Figure 1C). Next, they move to a clinical-grade system where they can observe a more pronounced difference, again by measuring water content in the setting of increased glycogen (thus an indirect measurement). They further test the system by developing a model of human muscle by using minced ground beef and adding increasing concentrations of glycogen. In this system, they note a steady increase in signal, with a clearly separable peak when the concentration of glycogen increases to 7%.

With the preclinical data above, Tan and colleagues then move to test the MSOT technology in a small clinical trial of 10 patients with LOPD and 10 healthy control volunteers. The trial included MSOT, muscle MRI, muscle ultrasound, and physical therapy functional testing. Healthy controls were age and sex matched. Healthy controls were heavier (73.9k g vs 65 kg) and taller (176.2 vs 172.3 cm). Functional testing with the R-PACT, QMFT, 6MWT, and TUAG were consistent with typical scores seen in LOPD. Strength on the MRC scale was consistent with proximal > distal weakness typically observed in LOPD.

In Figure 3, results from muscle MRI and ultrasound are given and are consistent with results obtained in clinic and previous reports in LOPD. There is visual difference on MRI and conventional ultrasound with increased fibrofatty replacement, but this is not significant using the elliptic or polygonal ROIs selected by the investigators.

Response 1

We thank the reviewer for the detailed analysis of our manuscript.

Comment 2

What are the noteworthy results?

Tan et al. present MSOT as a new, non-invasive imaging technique as a biomarker in Pompe disease. They test the technique in a glycogen and water mixture and a phantom muscle model with additional glycogen added (minced ground beef with increasing normal glycogen). They find that they cannot directly measure glycogen with their MSOT technique, but they can measure altered water signal in the presence of glycogen.

In a clinical trial, they claim superiority to muscle MRI and muscle ultrasound, but from a practical level appear to show at least non-inferiority to these imaging modalities.

The MSOT technique has advantages compared to MRI – it does not require sedation for

younger patients (though, note this study is in adult participants with mean age of 40 and not children).

The technique is faster than MRI or ultrasound as it can be performed in 10-20 seconds of imaging time.

Response 2

Thank you for bringing up these important points. The reviewer is certainly right that the question of superiority and/or inferiority cannot be clearly addressed in this study. The design, in particular the number of patients, was primarily to establish the methodology and to understand the imaging target down to a tissue level. A different study design would be necessary to validate non-inferiority/superiority (especially with regard to a larger sample size).

In addition, this reviewer is also correct regarding the age of our study cohort. Currently, there is only a single MSOT/optoacoustic imaging device with a valid CE-certification. This CE marking does imply its use in subjects older than 18 years of age, so that this study could only be realized in adult participants.

One major advantage of MSOT is, that is does not require sedation for performing imaging studies. This is only one example of the logistical superiority compared to MRI. From the perspective of everyday clinical practice, this advantage is generally relevant. In addition, this is a relevant advantage in all patients with an increased risk for sedation due to impaired respiratory function.

Comment 3

Will the work be of significance to the field and related fields? How does it compare to the established literature? If the work is not original, please provide relevant references.

The work is original for glycogen storage disease. Similar work has been performed by coauthors on this paper for other neuromuscular disorders.

Response 3

We would like to thank the reviewer for his correct and valuable assessment. To our knowledge, there is no comparable work in glycogen storage diseases.

Comment 4

Does the work support the conclusions and claims, or is additional evidence needed?

The claim appears valid that MSOT can distinguish LOPD from healthy volunteers, at least in this small study of 10 patients and 10 controls.

The authors claim that there are few biomarkers and these biomarkers can be unreliable -- "Follow-up and monitoring are ensured by clinical and functional tests, which are essentially dependent on the individual patient's active cooperation and performance.¹" however, they do not discuss other blood or urine based biomarkers, such as urine hex4, which is reliable and not influenced by the patient's cooperation of performance. Further discussion using updated guidelines for monitoring is recommended in the introduction and discussion.

Response 4

Thank you for this important suggestion. We agree with the reviewers' assessment and included an additional paragraph in the introduction.

The addition reads as follows in the introduction:

(...) The diagnosis of PD is usually established by confirmation of GAA deficiency, and confirmed by genetic testing.^{16,17} Furthermore, PD patients require regular clinical follow-up monitoring, especially to assess the response to ERT.^{8,9,17-20} While rapid determination of GAA in dried blood spots is possible, enzymatic analysis is unable to discriminate between patients with PD and those individuals harboring pseudo deficiency mutations. In this regard a tetraglucose oligomer (Glc(4)) in the urine and maltotetraose (Hex4) in plasma may hold promise as a biomarker to identify PD patients from individuals harboring pseudo deficiency mutations²¹ and even to assess response to ERT^{22,23}. However, the interpretation of the values is not trivial and must be considered with respect to the individual age of the patient¹⁸. To date, this is mostly ensured by clinical and functional tests, which are essentially dependent on the individual patient's active cooperation and performance.¹ (...)

Furthermore, the discussion was modified and additional references (guidelines/recommendations) were included.

Comment 5

The authors claim improved detection of disease compared to muscle MRI and ultrasound, however the correlation matrix in Figure 4 does not suggest a strong correlation between MSOT results and the various biomarkers and scores from the tests performed. It is also difficult to tell over the cohort if mild, moderate, and severe LOPD can be distinguished with MSOT. Will this biomarker be useful to trend overtime in an individual patient or does it just confirm disease is present? More sensitive and specific diagnostic tests already exist for Pompe Disease (enzyme activity testing and sequencing of GAA are widely available clinically and well validated).

Response 5

We thank the reviewer for the opportunity to emphasize the strengths of MSOT in this particular subject. As the reviewer correctly noted, the correlation matrix in Fig. 4k does not implicate a strong correlation between the MSOT results and the various biomarkers and scores of the tests performed. Nevertheless, MSOTcol and MSOTlip are both parameters that show a better correlation with pulmonary function tests and QMFT compared to validated imaging techniques being ultrasound and MRI. In this regard, we agree with the reviewer that MSOT might have potential to detect PD more sensitively compared to muscle MRI and ultrasound, but that further studies need to be conducted to improve the diagnostic value of MSOT.

We particularly like to appreciate the reviewer's suggestion to investigate the severity of PD with MSOT. To demonstrate this, we conducted further analyses (**Fig. 4i+j, Supplementary Figure 5**) by categorizing our participants into severity groups based on their QMFT results. Our cutoff for severely affected PD was a QMFT of 32 or half the possible score, for intermediately affected PD a QMFT of 48 or three quarters the possible score, and mildly affected as every QMFT score above 48. In this constellation, we observe an overall increase in MSOTlip and decrease in MSOTcol. Although we are aware that the interpretation is only possible considering the transversal study design and the small patient cohort, this result may be consistent with the fibro-fatty degeneration of PD muscles.

Fig. 4i:

Fig. 4j:

Supplementary Figure 5:

Furthermore, we agree with the reviewer that a biomarker to monitor disease progression is relevant for appropriate patient care, especially given the limited diagnostic tools and current treatment options. Although our transverse study design limits us in interpreting the longitudinal outlook, the results of previous work on optoacoustic imaging suggest that MSOT is able to sensitively monitor disease progression. We strongly agree with the reviewer that longitudinal

studies need to be conducted to explicitly demonstrate MSOT as a tool to monitor disease activity and therapy effect.

Comment 6

The claim in the discussion that this MSOT will serve as a substitute for invasive procedures (biopsy) appears overstated. The MSOT presented here is not measuring glycogen, thus there may be a number of different conditions which alter water signal in MSOT – the results may be more related to fibrofatty replacement than glycogen accumulation. This will not replace biopsy for challenging diagnostic cases where staining for glycogen and characterization by light and electron microscopy give vastly more information than this technique.

Response 6

We thank the reviewer for pointing this out. The reviewer correctly noted that glycogen has a high water binding capacity that can be measured with MSOT. Nevertheless, we supplemented our clinical results with preclinical experimental phantoms that indicate a glycogen-related optoacoustic signal alteration and not just fibrous-fatty degeneration of muscles. We agree with the reviewer that more studies need to be conducted and clinical experience gained for MSOT to be capable to replace highly-invasive muscle biopsy. However, we believe that MSOT has great potential to become a sensitive imaging modality in the future.

Ultimately, our do not aim to replace a possible biopsy with a different information content – as noted correctly. We more clearly stated its comparison to other imaging modalities (especially MRI). We have modified the text accordingly.

Comment 7

Are there any flaws in the data analysis, interpretation and conclusions? - Do these prohibit publication or require revision?

A concern is that, though MSOT is claimed to be “molecularly-sensitive,” the data presented here show an indirect measurement of glycogen accumulation through alteration in water content. This seems to fall short of the direct measurement of glycogen, that the benefits of a molecularly-sensitive technique claims. There may be a number of alterations in patients with Pompe Disease or other neuromuscular disorders which alter water content or fatty replacement of muscle that is unrelated to glycogen accumulation. The major advance would be direct measurement of glycogen in muscle, which is not proven in this paper with this MSOT method.

Response 7

We would like to thank the reviewer for this in-depth comment. We fully agree with the reviewer that it would be a major advance to measure glycogen in muscle directly. Nevertheless, to our knowledge this is the first study ever to investigate the optoacoustic behavior of glycogen in tissue. The primary aim of this study was therefore to test the biological validity of MSOT in glycogen storage disorder, establish the appropriate methodology and to understand the properties of the targeted tissue.

The degenerative muscle changes in LOPD include glycogen-related fibrous-fatty replacement of the muscle. In comparison, IOPD is known to show hardly any fibrous-fatty infiltration of the muscle in biopsies. We therefore believe that further studies are needed to better understand the glycogen-sensitive visualization and quantification of Pompe diseased muscle. In this regard, we believe that it might be helpful to image IOPD muscles in the future to better assess the glycogen effect in MSOT. We furthermore see potential in improving unmixing methods that could be complemented with the help of AI algorithms.

Thanks to this thorough suggestion, we have changed the wording “molecular-sensitive” accordingly and added the comments above in our discussion.

Comment 8

I would recommend revision for the language around biomarkers that includes urine and blood biomarkers (Hex4, CK, etc).

Response 8

Thank you for the valuable recommendation. We have added the suggested content to the manuscript and revised the wording as proposed. See also response 4.

Comment 9

From the data presented, I am not convinced that MSOT is superior to muscle MRI or ultrasound. It may be non-inferior. It would be helpful to visualize the results to show individual level data and separate by LOPD severity. Can severe LOPD be separated from moderate and mild LOPD? If there is no difference, how will this technique be helpful in a gene therapy or ERT trial? Both of these therapies slow, but do not fully eradicate, glycogen accumulation. Thus the patients will not go from diseased to normal. It is therefore critical for a biomarker to distinguish not only between healthy and disease, but between mild, moderate, and severe disease.

Response 9

We thank the reviewer for these important points. As mentioned in Response 2 the reviewer is correct. Due to the study design, neither superiority nor inferiority can be clearly addressed in this study. It is known in literature that MRI scans of the biceps in patients with LOPD hardly show any muscle involvement. We were able to confirm this morphologically and quantitatively with our MRI device. Nevertheless, MSOT showed statistically significant differences in the LOPD group compared to HV, which at least suggests the ability of MSOT to differentiate between patients and healthy subjects when it comes to the biceps muscle.

Furthermore, we agree with the reviewer that ERT only slows down the disease process and that it does not offer a cure.

However, the unmet need for objective biomarkers is a daily problem in the clinical practice of neuromuscular diseases. There are no validated parameters that can be used to classify severity or suggest the start of enzyme therapy. This study served to establish a methodology and aims to establish a biomarker in the future. We are of the opinion that the availability of a biomarker in disease monitoring is necessary for the complex care of patients. As the reviewer says, we are also interested in having more studies performed and experience obtained in order to offer a severity classification according to optoacoustic values in the future.

Comment 10

Is the methodology sound? Does the work meet the expected standards in your field?

The methodology appears reasonable for the development of a preclinical model and for the clinical trial with appropriate functional testing and current imaging modalities (MRI and conventional ultrasound).

Response 10

Thank you for your appreciation.

Comment 11

Is there enough detail provided in the methods for the work to be reproduced?

Yes, the methods are appropriately detailed.

Response 11

We thank the reviewer for the assessment.

Reviewer #2 (Remarks to the Author):

Comment 1

In this manuscript, the authors applied multispectral optoacoustic tomography (MSOT) to image biceps muscles for late-onset Pompe disease patients. Their results demonstrated that MSOT enabled imaging of subcellular disease pathology with increases in glycogen/water, collagen and lipid signals providing higher sensitivity to detect muscle degeneration than current clinical and imaging methods. Overall, the work is very interesting, which would help to advance the translation of the new imaging technique MSOT. In addition, the manuscript is well presented with proper results and discussion, and the limitations have also been stated. Therefore, I would recommend its publication after the following minor issue is addressed:

In the last paragraph of the Discussion section, it is stated “MSOT holds great potential to become a sensitive biomarker for diagnosing, phenotyping and monitoring patients with LOPD”.

The technique MSOT itself cannot be a biomarker, but it can serve as an operable technique for diagnosing, phenotyping and monitoring patients with LOPD.

Hence, This statement should be modified.

Response 1

We thank the reviewer for this suggestion. As proposed the statement in the discussion was modified and reads as follows:

“(...) In conclusion, MSOT holds great potential to become a sensitive imaging technology for diagnosing, phenotyping and monitoring patients with LOPD. (...)”

Reviewer #3 (Remarks to the Author):

Comment 1

This referee assessed only the statistical part and report of results. Regarding these sections I have not particular comments. The analyses were simple and results adequately reported.

I have only a question. Is it possible to see the results in a multivariable analysis or isn't feasible.

Response 1

Thank you for this excellent and important suggestion. As the reviewer correctly assumed, we considered various statistical analyses. We discussed performing paired MANOVA, but due to the exploratory character of our study and the lack of theoretical model for the parameters analyzed, we realized that the clustering of variables would not be based on a theoretical background. And to include all variables in one model, the sample size was not sufficient. Additionally, research (Tabachnik and Fidell (2012) and Field (2024)) recommended against using a model without a theoretical background. Nonetheless, we agree with the reviewer and consider that in future studies, as more evidence for MSOT emerges, multivariate testing should be conducted to account for alpha-inflation.

However, in this pilot study, our aim was to explore individual parameters to determine singular or multiple variables that would allow us to differentiate between affected and healthy participants. This may now be applied in future work.

Tabachnick, B.G. and Fidell, L.S (2012), *Using Multivariate Statistics.* 6th Edition, Person Education, Boston

Field, A. (2024). *Discovering statistics using IBM SPSS Statistics.* SAGE Publications Limited.

REVIEWERS' COMMENTS

Reviewer #1 (Remarks to the Author)

Comment 1

In "Non-invasive optoacoustic imaging of glycogen-storage and muscle degeneration in Late-onset Pompe disease," Tan and colleagues investigate whether multispectral optoacoustic tomography (MSOT) can be applied to measure glycogen accumulation in the muscle of patients with Late-onset Pompe Disease. MSOT is a modified ultrasound-like approach which emits near-infrared light and records sound waves emitted. It is non-invasive and the wavelengths emitted can be customized to measure molecules of interest with some precision.

Tan and colleagues began by testing in an agarose and water system if glycogen could be measured with spectrophotometer. They do not identify any signal from glycogen itself, but they are able to detect an increase in the absorption of water, which changes when the concentration of glycogen increases. They thus appear to be using an indirect measurement of glycogen accumulation (Figure 1A,B). They then move onto a preclinical imaging system, where again they can measure a difference in water content as a surrogate for increased glycogen (Figure 1C). Next, they move to a clinical-grade system where they can observe a more pronounced difference, again by measuring water content in the setting of increased glycogen (thus an indirect measurement). They further test the system by developing a model of human muscle by using minced ground beef and adding increasing concentrations of glycogen. In this system, they note a steady increase in signal, with a clearly separable peak when the concentration of glycogen increases to 7%.

With the preclinical data above, Tan and colleagues then move to test the MSOT technology in a small clinical trial of 10 patients with LOPD and 10 healthy control volunteers. The trial included MSOT, muscle MRI, muscle ultrasound, and physical therapy functional testing. Healthy controls were age and sex matched. Healthy controls were heavier (73.9k g vs 65 kg) and taller (176.2 vs 172.3 cm). Functional testing with the R-PACT, QMFT, 6MWT, and TUAG were consistent with typical scores seen in LOPD. Strength on the MRC scale was consistent with proximal > distal weakness typically observed in LOPD.

In Figure 3, results from muscle MRI and ultrasound are given and are consistent with results obtained in clinic and previous reports in LOPD. There is visual difference on MRI and conventional ultrasound with increased fibrofatty replacement, but this is not significant using the elliptic or polygonal ROIs selected by the investigators.

Response 1

We thank the reviewer for the detailed analysis of our manuscript.

Comment 2

What are the noteworthy results?

Tan et al. present MSOT as a new, non-invasive imaging technique as a biomarker in Pompe disease. They test the technique in a glycogen and water mixture and a phantom muscle model with additional glycogen added (minced ground beef with increasing normal glycogen). They find that they cannot directly measure glycogen with their MSOT technique, but they can measure altered water signal in the presence of glycogen.

In a clinical trial, they claim superiority to muscle MRI and muscle ultrasound, but from a practical level appear to show at least non-inferiority to these imaging modalities.

The MSOT technique has advantages compared to MRI – it does not require sedation for younger patients (though, note this study is in adult participants with mean age of 40 and not children).

The technique is faster than MRI or ultrasound as it can be performed in 10-20 seconds of imaging time.

Response 2

Thank you for bringing up these important points. The reviewer is certainly right that the question of superiority and/or inferiority cannot be clearly addressed in this study. The design, in particular the number of patients, was primarily to establish the methodology and to understand the imaging target down to a tissue level. A different study design would be necessary to validate non-inferiority/superiority (especially with regard to a larger sample size).

In addition, this reviewer is also correct regarding the age of our study cohort. Currently, there is only a single MSOT/optoacoustic imaging device with a valid CE-certification. This CE marking does imply its use in subjects older than 18 years of age, so that this study could only be realized in adult participants.

One major advantage of MSOT is, that it does not require sedation for performing imaging studies. This is only one example of the logistical superiority compared to MRI. From the perspective of everyday clinical practice, this advantage is generally relevant. In addition, this is a relevant advantage in all patients with an increased risk for sedation due to impaired respiratory function.

Reviewer Response to comment 2:

I accept with the responses. The article does not state "superiority." It would be interesting to include pediatric patients in a follow up study, given that LOPD can present in early childhood or even infancy.

Comment 3

Will the work be of significance to the field and related fields? How does it compare to the established literature? If the work is not original, please provide relevant references.

The work is original for glycogen storage disease. Similar work has been performed by coauthors on this paper for other neuromuscular disorders.

Response 3

We would like to thank the reviewer for his correct and valuable assessment. To our knowledge, there is no comparable work in glycogen storage diseases.

Reviewer Response to comment 3:

Agreed.

Comment 4

Does the work support the conclusions and claims, or is additional evidence needed?

The claim appears valid that MSOT can distinguish LOPD from healthy volunteers, at least in this small study of 10 patients and 10 controls.

The authors claim that there are few biomarkers and these biomarkers can be unreliable --"Follow-up and monitoring are ensured by clinical and functional tests, which are essentially dependent on the individual patient's active cooperation and performance.1," however, they do not discuss other blood or urine based biomarkers, such as urine hex4, which is reliable and not influenced by the patient's cooperation or performance. Further discussion using updated guidelines for monitoring is recommended in the introduction and discussion.

Response 4

Thank you for this important suggestion. We agree with the reviewers' assessment and included an additional paragraph in the introduction.

The addition reads as follows in the introduction:

(...) The diagnosis of PD is usually established by confirmation of GAA deficiency, and confirmed by genetic testing.^{16,17} Furthermore, PD patients require regular clinical follow-up monitoring, especially to assess the response to ERT.^{8,9,17-20} While rapid determination of GAA in dried blood spots is possible, enzymatic analysis is unable to discriminate between patients with PD and those individuals harboring pseudo deficiency mutations. In this regard a tetraglucose oligomer (Glc(4)) in the urine and maltotetraose (Hex4) in plasma may hold promise as a biomarker to identify PD patients from individuals harboring pseudo deficiency mutations²¹ and even to assess response to ERT^{22,23}. However, the interpretation of the values is not trivial and must be considered with respect to the individual age of the patient¹⁸. To date, this is mostly ensured by clinical and functional tests, which are essentially dependent on the individual patient's active cooperation and performance.¹ (...)

Furthermore, the discussion was modified and additional references (guidelines/recommendations) were included.

Reviewer Response to comment 4:

I appreciate the edits in the introduction and the now more nuanced discussion.

Comment 5

The authors claim improved detection of disease compared to muscle MRI and ultrasound, however the correlation matrix in Figure 4 does not suggest a strong correlation between MSOT results and the various biomarkers and scores from the tests performed. It is also difficult to tell over the cohort if mild, moderate, and severe LOPD can be distinguished with MSOT. Will this biomarker be useful to trend overtime in an individual patient or does it just confirm disease is present? More sensitive and specific diagnostic tests already exist for Pompe Disease (enzyme activity testing and sequencing of GAA are widely available clinically and well validated).

Response 5

We thank the reviewer for the opportunity to emphasize the strengths of MSOT in this particular subject. As the reviewer correctly noted, the correlation matrix in Fig. 4k does not implicate a strong correlation between the MSOT results and the various biomarkers and scores of the tests performed. Nevertheless, MSOTcol and MSOTlip are both parameters that show a better correlation with pulmonary function tests and QMFT compared to validated imaging techniques being ultrasound and MRI. In this regard, we agree with the reviewer that MSOT might have potential to detect PD more sensitively compared to muscle MRI and ultrasound, but that further studies need to be conducted to improve the diagnostic value of MSOT.

We particularly like to appreciate the reviewer's suggestion to investigate the severity of PD with MSOT. To demonstrate this, we conducted further analyses (Fig. 4i+j, Supplementary Figure 5) by categorizing our participants into severity groups based on their QMFT results. Our cutoff for severely affected PD was a QMFT of 32 or half the possible score, for intermediately affected PD a QMFT of 48 or three quarters the possible score, and mildly affected as every QMFT score above 48. In this constellation, we observe an overall increase in MSOTlip and decrease in MSOTcol. Although we are aware that the interpretation is only possible considering the transversal study design and the small patient cohort, this result may be consistent with the fibro-fatty degeneration of PD muscles.

Fig. 4i:

Fig. 4j:

Supplementary Figure 5:

Furthermore, we agree with the reviewer that a biomarker to monitor disease progression is relevant for appropriate patient care, especially given the limited diagnostic tools and current treatment options. Although our transverse study design limits us in interpreting the longitudinal outlook, the results of previous work on optoacoustic imaging suggest that MSOT is able to sensitively monitor disease progression. We strongly agree with the reviewer that longitudinal studies need to be conducted to explicitly demonstrate MSOT as a tool to monitor disease activity and therapy effect.

Reviewer Response to comment 4:

I appreciate the additional effort to analyze subgroups. Given the small sample size, it is not surprising that they do not meet statistical significance, but the trends do appear to move in the expected direction. I look forward to future studies with larger cohorts followed over time to verify these findings.

Comment 6

The claim in the discussion that this MSOT will serve as a substitute for invasive procedures (biopsy) appears overstated. The MSOT presented here is not measuring glycogen, thus there may be a number of different conditions which alter water signal in MSOT – the results may be more related to fibrofatty replacement than glycogen accumulation. This will not replace biopsy for challenging diagnostic cases where staining for glycogen and characterization by light and electron microscopy give vastly more information than this technique.

Response 6

We thank the reviewer for pointing this out. The reviewer correctly noted that glycogen has a high water binding capacity that can be measured with MSOT. Nevertheless, we supplemented our clinical results with preclinical experimental phantoms that indicate a glycogen-related optoacoustic signal alteration and not just fibrous-fatty degeneration of muscles. We agree with the reviewer that more studies need to be conducted and clinical experience gained for MSOT to be capable to replace highly-invasive muscle biopsy. However, we believe that MSOT has great potential to become a sensitive imaging modality in the future.

Ultimately, our do not aim to replace a possible biopsy with a different information content – as noted correctly. We more clearly stated its comparison to other imaging modalities (especially MRI). We have modified the text accordingly.

Reviewer Response to comment 6:

I accept the authors' argument.

Comment 7

Are there any flaws in the data analysis, interpretation and conclusions? - Do these prohibit publication or require revision?

A concern is that, though MSOT is claimed to be "molecularly-sensitive," the data presented here show an indirect measurement of glycogen accumulation through alteration in water content. This seems to fall short of the direct measurement of glycogen, that the benefits of a molecularly-sensitive technique claims. There may be a number of alterations in patients with Pompe Disease or other neuromuscular disorders which alter water content or fatty replacement of muscle that is unrelated to glycogen accumulation. The major advance would be direct measurement of glycogen in muscle, which is not proven in this paper with this MSOT method.

Response 7

We would like to thank the reviewer for this in-depth comment. We fully agree with the reviewer that it would be a major advance to measure glycogen in muscle directly. Nevertheless, to our knowledge this is the first study ever to investigate the optoacoustic behavior of glycogen in tissue. The primary aim of this study was therefore to test the biological validity of MSOT in glycogen storage disorder, establish the appropriate methodology and to understand the properties of the targeted tissue.

The degenerative muscle changes in LOPD include glycogen-related fibrous-fatty replacement of the muscle. In comparison, IOPD is known to show hardly any fibrous-fatty infiltration of the muscle in biopsies. We therefore believe that further studies are needed to better understand the glycogen-sensitive visualization and quantification of Pompe diseased muscle. In this regard, we believe that it might be helpful to image IOPD muscles in the future to better assess the glycogen effect in MSOT. We furthermore see potential in improving unmixing methods that could be complemented with the help of AI algorithms.

Thanks to this thorough suggestion, we have changed the wording "molecular-sensitive" accordingly and added the comments above in our discussion.

Reviewer Response to comment 7:

I appreciate the wording adjustment. I look forward to further studies in IOPD or other glycogen storage diseases where this technique may serve as a valuable tool.

Comment 8

I would recommend revision for the language around biomarkers that includes urine and blood biomarkers (Hex4, CK, etc).

Response 8

Thank you for the valuable recommendation. We have added the suggested content to the manuscript and revised the wording as proposed. See also response 4.

Reviewer Response to comment 8:

Accepted.

Comment 9

From the data presented, I am not convinced that MSOT is superior to muscle MRI or ultrasound. It may be non-inferior. It would be helpful to visualize the results to show individual level data and separate by LOPD severity. Can severe LOPD be separated from moderate and mild LOPD? If there is no difference, how will this technique be helpful in a gene therapy or ERT trial? Both of these therapies slow, but do not fully eradicate, glycogen accumulation. Thus the patients will not go from diseased to normal. It is therefore critical for a biomarker to distinguish not only between healthy and disease, but between mild, moderate, and severe disease.

Response 9

We thank the reviewer for these important points. As mentioned in Response 2 the reviewer is correct. Due to the study design, neither superiority nor inferiority can be clearly addressed in this study. It is known in literature that MRI scans of the biceps in patients with LOPD hardly show any muscle involvement. We were able to confirm this morphologically and quantitatively with our MRI device. Nevertheless, MSOT showed statistically significant differences in the LOPD group compared to HV, which at least suggests the ability of MSOT to differentiate between patients and healthy subjects when it comes to the biceps muscle.

Furthermore, we agree with the reviewer that ERT only slows down the disease process and that it does not offer a cure.

However, the unmet need for objective biomarkers is a daily problem in the clinical practice of neuromuscular diseases. There are no validated parameters that can be used to classify severity or suggest the start of enzyme therapy. This study served to establish a methodology and aims to

establish a biomarker in the future. We are of the opinion that the availability of a biomarker in disease monitoring is necessary for the complex care of patients. As the reviewer says, we are also interested in having more studies performed and experience obtained in order to offer a severity classification according to optoacoustic values in the future.

Reviewer Response to comment 9:

I look forward to seeing further work with MSOT. I am encouraged by the work in this small cohort, described in an earlier comment, to separate patients into severity groups. In the future, it may also be beneficial to obtain data from multiple muscle groups, given how quick the imaging time is with MSOT.

Reviewer #2 (Remarks to the Author):

The manuscript has been improved and is ready for publication.

Reviewer #3 (Remarks to the Author):

I am agree with the authors on applicability of Multivariable analyses on this sample.
I have not further comments.